# Forming cognitive maps for abstract spaces: the roles of the human hippocampus and orbitofrontal cortex
Yidan Qiu [1], Huakang Li [2], Jiajun Liao [1], Kemeng Chen [1], Xiaoyan Wu [1], Bingyi Liu [1] & Ruiwang Huang [1] ✉

How does the human brain construct cognitive maps for decision-making and inference? Here, we conduct an fMRI study on a navigation task in multidimensional abstract spaces. Using a deep neural network model, we assess learning levels and categorized paths into exploration and exploitation stages. Univariate analyses show higher activation in the bilateral hippocampus and lateral prefrontal cortex during exploration, positively associated with learning level and response accuracy. Conversely, the bilateral orbitofrontal cortex (OFC) and retrosplenial cortex show higher activation during exploitation, negatively associated with learning level and response accuracy. Representational similarity analysis show that the hippocampus, entorhinal cortex, and OFC more accurately represent destinations in exploitation than exploration stages. These findings highlight the collaboration between the medial temporal lobe and prefrontal cortex in learning abstract space structures. The hippocampus may be involved in spatial memory formation and representation, while the OFC integrates sensory information for decision-making in multidimensional abstract spaces.

How do human brains build a cognitive map for guiding flexible behavior? Since cognitive maps were proposed by Tolman (1948)[1] from observing the foraging behavior of rats in mazes, they have been believed to support the flexible behavior of animals. For humans, a cognitive map refers to a systematic organization of knowledge[2]. It has not only been used to explain people's spatial navigation in physical spaces[3–5] but has also been generalized to nonphysical spaces or abstract spaces to explain the flexible thinking and behavior of humans[2,6,7]. Recent studies found that the medial temporal lobe (MTL), especially the hippocampal-entorhinal (HIP-EC) system, and the orbitofrontal cortex (OFC), were the major brain regions supporting the use of cognitive maps for both physical and abstract spaces[4,8–10].

Previous studies suggested that the HIP-EC system is critical for forming a cognitive map. First, the HIP-EC system was proposed as a core region for developing a cognitive map by providing a mental representation of the spatial layout of an environment and the structure of a nonspatial item as well as a spatio-temporal framework for relevant experiences[11–13]. Second, the HIP was suggested to play a critical role in learning, spatial reasoning, and relational memory[12,14,15]. Previous studies[16,17] found that the HIP is engaged in dynamically updating the representation of objects and transmitting short-term memory into long-term memory[18,19]. The place cells in the HIP were found to map the relative locations in a physical space[13] and to

replay the spatio-temporal sequence to support the recollection and consolidation of newly acquired information[20–22]. In addition, different HIP activation patterns were found for different stages of exploring the environment. For example, ref. 23 showed distinct activation patterns in the HIP when humans tracked the distance to goal in highly familiar versus newly learned environments. Similarly, ref. 24 observed different representations in the HIP for visually similar but conceptually different stimuli after learning the task structure.

The OFC is another candidate for involvement in forming a cognitive map. Studies suggested that it encodes cognitive maps of the task space in humans for planning complex behaviors and for making inferences when the state was not directly observable[25–27]. The various functions attributed to the OFC, such as value prediction, credit assignment, response inhibition, and emotions, may stem from its representation of cognitive maps[27–29]. The OFC receives inputs from the sensory areas, HIP, and striatum, suggesting its involvement in integrating sensory observations and forming associative representations[28,30]. Previous studies suggested that the OFC and HIP encode different aspects of cognitive maps[29]. However, it remains unclear how the two brain regions collaborate to support flexible behavior.

The current study aimed to understand whether the HIP, EC, and OFC build an internal representation when people are exploring an environment

[1]School of Psychology; Center for the Study of Applied Psychology; Key Laboratory of Mental Health and Cognitive Science of Guangdong Province; Key Laboratory of Brain, Cognition and Education Sciences, Ministry of Education; South China Normal University, Guangzhou 510631, China. [2]School of Computer Science and Engineering, South China University of Technology, Guangzhou 510006, China. ✉e-mail: ruiwang.huang@gmail.com

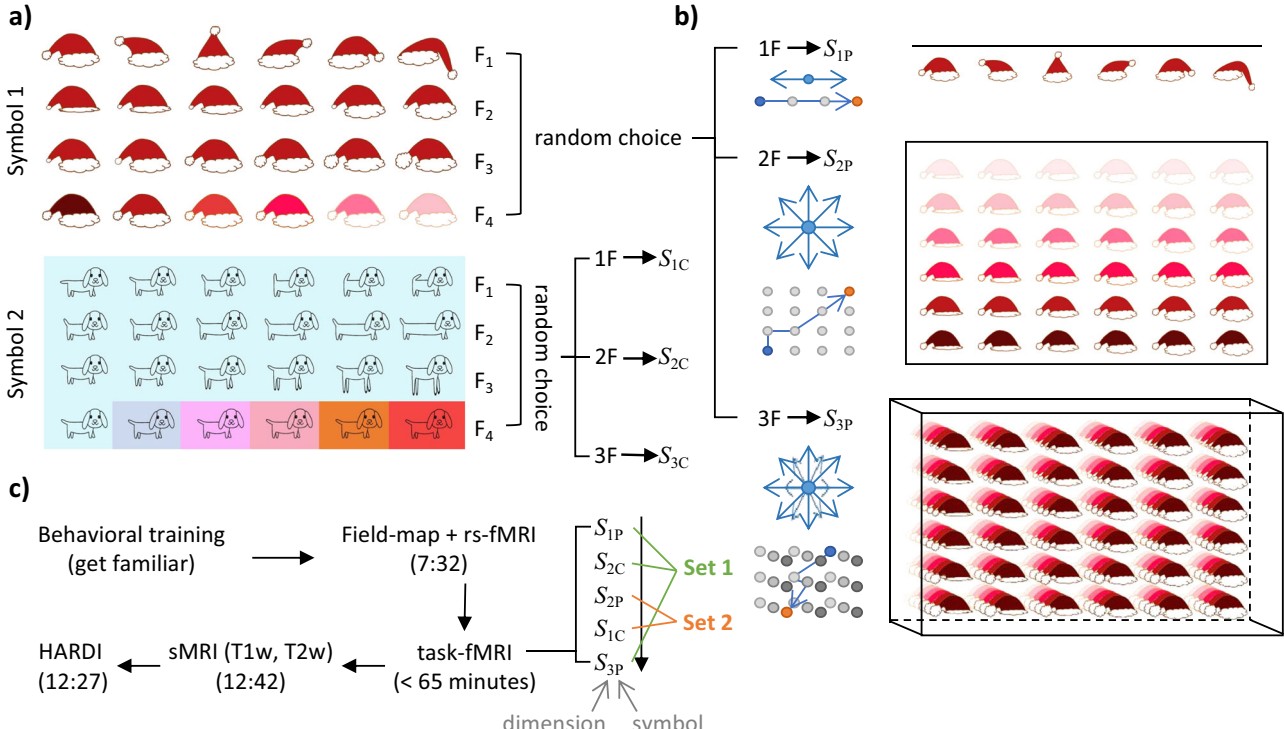

**Fig. 1 | Schematic of the construction of the abstract spaces and the fMRI scanning procedure. a** The two basic symbols and their four features ($F_1$, $F_2$, $F_3$, and $F_4$). A certain feature or a dimension consisted of six discrete values. The four features of the hat symbol were tilt angle, brim width, pompom size, and color lightness. The four features of the dog symbol were tail direction, body length, leg length, and color tone. The symbols and features were created by the authors. **b** Example of the 1D, 2D, and 3D abstract spaces constructed with one feature (1 F), two features (2 F), and three features (3 F) of the hat symbol. In the example, the hat was taken as the primary symbol (denoted as P) to construct three different dimensional abstract spaces, the $S_{1P}$, $S_{2P}$, and $S_{3P}$. The other symbol, dog, was taken as the control symbol (denoted as C) to construct another three abstract spaces, the $S_{1C}$, $S_{2C}$, and $S_{3C}$ (the $S_{3C}$ was not used in the fMRI experiment). If the dog was taken as the primary symbol, the $S_{1P}$, $S_{2P}$, and $S_{3P}$ would be constructed using the dog symbol, and the $S_{1C}$, $S_{2C}$, and $S_{3C}$ would be constructed using the hat symbol. Each abstract space used in the navigation task for a subject was constructed by randomly choosing 1–3 features of a specific symbol. The compasses show the directions that the subjects could take to move in the abstract space during the experiment. The circles and arrows below

the compasses are examples of a shortest path from a starting point (blue circle) to a destination (orange circle) in the 1D, 2D, and 3D abstract spaces. Each dot represents a location in the abstract space. More than one shortest path may exist between a certain current and goal locations (Fig. S3, Supplementary Information), so the arrow indicates one of the shortest paths. **c** The procedure of the experiment. During the fMRI scanning, the subjects performed the navigation task in five different abstract spaces separately in a fixed order of $S_{1P}$, $S_{2C}$, $S_{2P}$, $S_{1C}$, and $S_{3P}$. The $S_{1P}$, $S_{2C}$, and $S_{3P}$ were collectively referred to as Set 1 because these three were the first space of each dimensionality presented to the subject. The $S_{1C}$ and $S_{2P}$ were collectively referred to as Set 2 because these two were the second space of each dimensionality presented to the subject. Half of the subjects used the hat as the primary symbol, and the other half of the subjects used the dog as the primary symbol. Abbreviations: D, dimension; F, feature; P, primary symbol; C, control symbol; $S_{1P}$, $S_{1C}$, $S_{2P}$, $S_{2C}$, $S_{3P}$, and $S_{3C}$ indicate six different multi-dimensional abstract spaces; Set 1 and Set 2 refer to the first and the second space of each dimensionality presented to the subject. rs-fMRI, resting-state fMRI; sMRI, structural magnetic resonance image; HARDI, high angular resolution diffusion-weighted imaging.

or learning a concept. When making decisions, we often need to integrate various factors, which means that the decision space could be multi-dimensional. For example, we only need to consider the size to know that a basketball is bigger than a tennis ball (1D), but when choosing a school for our children, we may consider factors such as the school's ranking, history, and geographical location (3D). To identify the brain regions involved in constructing cognitive maps for multidimensional abstract spaces, we designed navigation tasks in 1D, 2D, and 3D abstract spaces (Fig. 1) and acquired fMRI data while the subjects were performing the tasks in abstract spaces (Fig. 2). To capture the different brain activations and representations that occur while learning in the abstract spaces, we separated the navigation paths into two stages, the exploration and exploitation stages, according to each subject's behavioral performance (response accuracy and response time). Considering the differences in behavior performance between exploration and exploitation are subtle, we applied a deep neural network (DNN) to estimate how much the subjects have learned the structure of abstract spaces (learning level) during each navigation path. Then, a $k$-means algorithm was used to separate the navigation into exploration and exploitation stages (Fig. 3). By analyzing task-fMRI data, we compared the brain activations and representation patterns between the two stages while navigating in abstract spaces. If the HIP, EC, and OFC show

different activation strengths and representation patterns between the two learning stages, and if the differences were correlated to the learning level or behavioral performance, then they are involved in constructing cognitive maps for the abstract spaces.

## Methods

### Subjects

Twenty-seven healthy adult subjects (14 women) took part in this experiment (mean age, 21.78 years; range, 18−29 years). All had normal or corrected-to-normal vision. None of them had any history of neurological disease or brain disorders. All of the subjects finished five task-fMRI scans within a single session. The data from two subjects were excluded due to their poor task performance during fMRI scanning. The study was approved by the Institutional Review Board (IRB) of the SCNU (# 2019-3-062). All ethical regulations relevant to human research participants were followed. Each subject gave written informed consent prior to the study and was compensated for participation after the experiment.

### Experimental design and procedures

The experiment lasted two days (Fig. S1, Supplementary Information). On Day 1, we performed the following items for each subject outside the

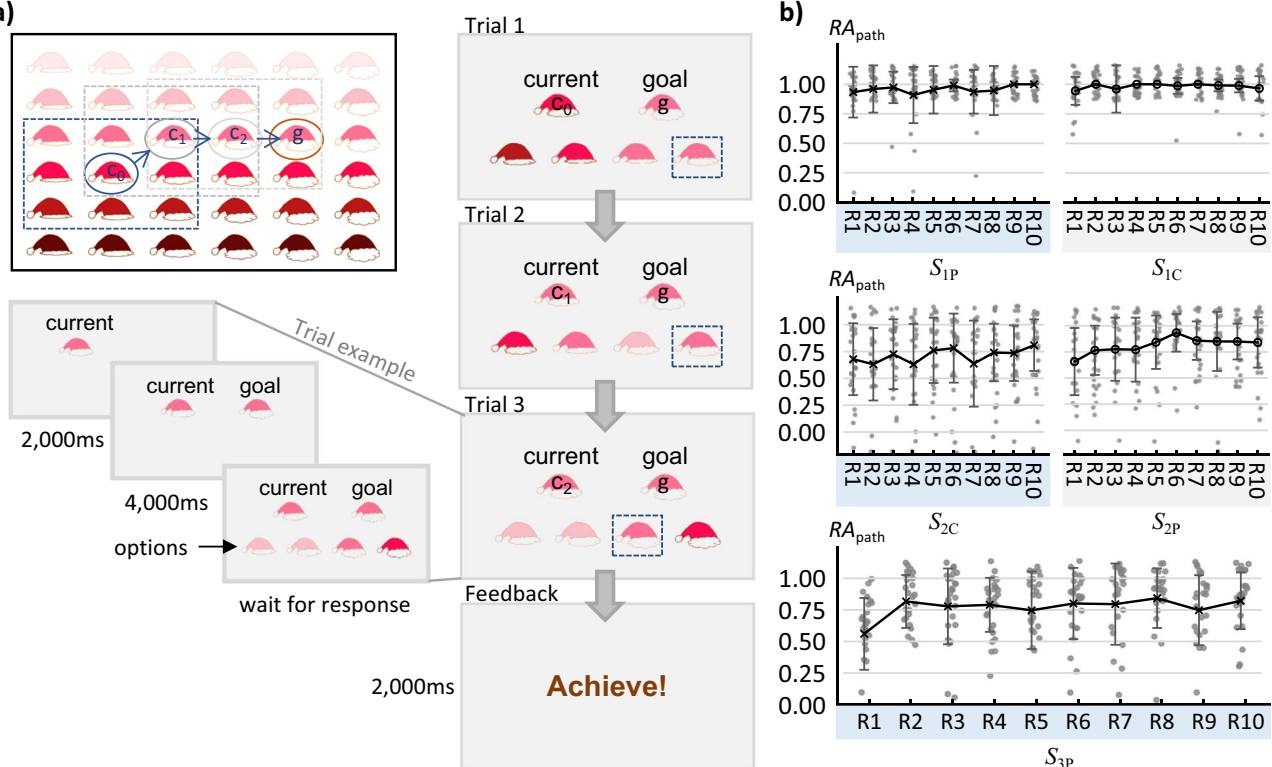

**Fig. 2 | An example of the navigation task in a 2D abstract space for a subject and the response accuracy of the paths ($RA_{path}$) in the five abstract spaces. a** Procedure of the navigation task. In each path (from $c_0$ to $g$), two locations (specific combination of features described in Fig. 1) in the abstract space were chosen randomly as the starting point (also as the first current location) and the destination. The subjects navigated to the destination from the starting point by choosing the given options. With an optimal option, the subject can reach to the destination with the lowest number of steps. In each trial, the options were chosen randomly around the current location (within the dotted frame). For a subject, if the chosen option was not same as the destination, the chosen option became the current location of the next trial and the destination remained the same; otherwise, a feedback screen would indicate the completion of the path. Each trial started with a cue indicating the current location for 2 s, then showed the destination for 4 s, followed by the options, and ended with the subject's response. **b** Average response accuracy for each path ($RA_{path}$) across all the 25 subjects. The error bars indicate standard deviation. The underlying source data is supplied in Supplementary Data 1. Abbreviations: $c_n$, the current location of the $n^{th}$ step; g, goal location; $S_{1P}$, $S_{1C}$, $S_{2P}$, $S_{2C}$, and $S_{3P}$ represent the five abstract spaces; R1, R2, ..., and R10 indicate ten different paths or routes.

scanner: (1) collected basic demographic information, (2) evaluated the subject's ability with respect to direction, reasoning, working memory, discriminability, and rule learning, (3) measured the subject's perception of stress, and (4) let the subject familiarize with the navigation task in the abstract spaces designed for the behavioral training (Figs. S1 and S2, Supplementary Information). The detailed information about the experiment and materials used on Day 1 is described in Supplementary Note 1. The data from items (2) and (3) were not used in the current study. On Day 2, each subject was trained again outside the scanner to ensure that they fully understood the navigation task (Supplementary Note 2). Afterward, they attended the MRI scan when performing the navigation task in five different abstract spaces (Fig. 1).

For each subject, five abstract spaces were used in the task-fMRI experiment. Figure 1 illustrates the steps for the construction of these abstract spaces. Specifically, we first defined two basic symbols, hat and dog. For each subject, one of the basic symbols was defined as the primary symbol (denoted as P), which was used to construct three abstract spaces used in the fMRI experiment. The other basic symbol was defined as the control symbol (denoted as C), which was used to construct the remaining two abstract spaces used in the fMRI experiment. We counterbalanced the two basic symbols among the subjects; 13 of the subjects used hat as the primary symbol and the other 14 subjects used dog as the primary symbol.

We defined a dimension by manipulating a feature of a basic symbol (Fig. 1a). To avoid the potential effect of the fixed features used in the experiment on the results, we defined four features corresponded to four dimensions for each basic symbol. The number of dimensions in the abstract space was in a range from one to three, so no more than three

features of a given symbol would be used to construct an abstract space. For each subject, the features of each abstract space were randomly chosen from the four features. Figure 1a shows that the four features of hat were tilt angle, brim width, pompom size, and color lightness, and the four features of the dog were tail direction, body length, leg length, and color tone.

After the symbols and dimensions were determined, the five multi-dimensional abstract spaces were constructed with the corresponding number of features for a specific symbol (Fig. 1). These multidimensional spaces were denoted as follows (Fig. 1a, b). (1) $S_{1P}$: a 1D space of the primary symbol; (2) $S_{1C}$: another 1D space of the control symbol; (3) $S_{2P}$: a 2D space of the primary symbol; (4) $S_{2C}$: another 2D space of the control symbol; and (5) $S_{3P}$: a 3D space of the primary symbol. Specifically, when hat was the primary symbol, $S_{1P}$ was constructed with a feature chosen randomly from the four features of the hat, $S_{1C}$ with a feature chosen randomly from the four features of the dog, $S_{2P}$ with two features chosen randomly from the four features of the hat, and so on (Fig. 1b). The location of a point in the space was described as a number (in 1D) or a pair of numbers (in 2D) or a triplet of numbers (in 3D) that specified distances from the coordinate axes (Fig. S2, Supplementary Information).

During the fMRI scanning, the subjects were required to perform the navigation task in five abstract spaces in a fixed order of $S_{1P}$, $S_{2C}$, $S_{2P}$, $S_{1C}$, and $S_{3P}$. The spaces that appeared first for each dimension in the experimental order, $S_{1P}$, $S_{2C}$, and $S_{3P}$, were collectively referred to as Set 1. Similarly, the spaces that appeared second for each dimension, $S_{1C}$ and $S_{2P}$, were collectively referred to as Set 2 (Fig. 1c). To control the total scanning time, we set the maximum scanning time for each fMRI scan to 10 min (400 volumes) for the task in the 1D space and 15 min (600 volumes) for the task in each of

**Fig. 3 | Schematic and results for the separation of the navigation paths into the exploration and exploitation stages. a** Definition of the early learning (green), mid-learning (white), and late learning (orange). The first three paths and the last three paths of each space were labeled as the early and late learning phases, respectively. These labeled paths were used in training the deep neural network (DNN). **b** Construction of the DNN prediction model. The DNN contains 2 hidden layers between the input and output layers. The input data were a $N_{step}$-by-2 matrix $\mathbf{B} = [RA_1, RT_1;...;RA_{N_{step}}, RT_{N_{step}}]$ of a path, where $RA_{N_{step}}$ and $RT_{N_{step}}$ represent response accuracy and response time of the $N$th step, respectively. The values of the two units in the output layer indicate the probability of the path being categorized as early learning phase (denoted as $P(early)$) and being categorized as late learning phase (denoted as $P(late) = 1 - P(early)$). The $\mathbf{B}$ matrices of all paths were input to the trained DNN to obtain the $P(early)$. **c** Categorization the navigation paths into the exploration and exploitation stages. A $k$-means algorithm was used to categorize the paths into the exploration and exploitation stages based on the $P(early)$. **d** Performance of the DNN prediction model in the 25 subjects. During training, the DNN model predicted the label of the paths in the test set. The accuracy was calculated as the ratio of correctly predicted paths to the total paths in the testing set and averaged across the 100 iterations. The model accuracy value was significantly higher than the chance level ($0.25 = 0.5 / n_{test}$). **e** The number of paths categorized as exploitation stage by the $k$-means algorithm in each space in the 25 subjects. The error bars indicate standard deviation. The underlying source data is supplied in Supplementary Data 1. ***$p < .001$.

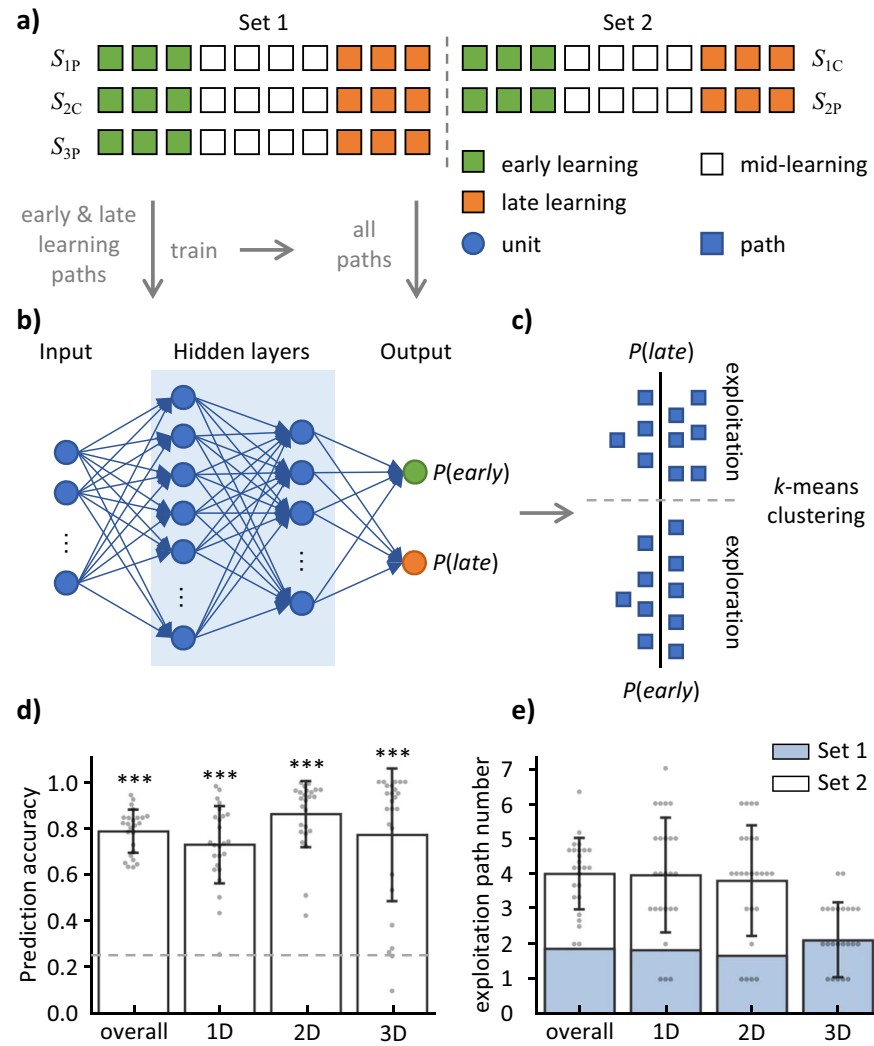

the 2D and 3D spaces. For a given task, the fMRI scan ended at the maximum time even if the subjects did not complete the task. Only 2 subjects failed to complete the navigation task within the maximum scanning time, one in a 1D and a 3D spaces, and the other in a 3D space. The data of these 2 subjects were excluded from the subsequent analyses. Detailed information of the scanning lengths is listed in Table S1 (Supplementary Information).

Figure 2a illustrates the procedure of the navigation task in an abstract space. The program was coded using PsychoPy (version 2021.2.3, https://www.psychopy.org/). The stimuli were displayed against a gray background. For the task in each abstract space, the subjects needed to navigate to 10 destinations separately from 10 different starting points. The trajectory that the subjects navigated from a starting point to a destination was defined as a path. In other words, each subject generated ten paths in each of the five abstract spaces. At the beginning of each path, two different locations were selected in the abstract space randomly, one as the starting point and the other one as the destination.

For each subject, a trial began with the presentation of a starting point or the current location and ended with the subject's selection, followed by an inter-trial interval (ITI) lasting 3−9 s (Fig. 2a). The current location along with a word, "current", was displayed at the top left of the screen, followed by the destination along with a word, "goal", displayed at the top right of the screen after 2 s. Four seconds later, the options were displayed at the bottom of the screen. These options were selected from locations around the current location (within the dotted frames in Fig. 2a) following the rules described in Supplementary Note 3. The subject pressed one of four buttons on an MRI-

compatible 4-button bimanual button-box to make the selection when the options were presented. The four options were arranged according to their locations in the abstract space relative to the current location, with the location below the current location as the first option, the location to the right below the current location as the last option, and the other locations arranged in a clockwise direction (Fig. S3, Supplementary Information). The first two options corresponded to the buttons on the left, and the other two options corresponded to the buttons on the right.

If the subject selected the option that was consistent with the goal location, the path was completed with a word "Achieved!" and a coin displayed on the screen as feedback, indicating that the path had ended. The subjects were told before the fMRI scans that they could obtain additional compensation based on the number of coins that they obtained in the task. Alternatively, if the subjects selected an option that did not correspond to the goal location, the path continued, and the next trial started after the ITI. In the new trial, the subjects' selection in the last trial became the current location and the goal remained the same as in the previous trial. These steps repeated until the subject selected an option that corresponded to the goal location. To avoid excessive time spent on a path, we implemented a 20% chance of terminating a path if the subject failed to select the optimal option. Once a path was terminated, the word "Break!" displayed on the screen, indicating the end of the path, and the subject could not get a coin from a terminated path. The terminated paths were also included in the following data analyses. The next path began after a fixation cross, which lasted for 6−13 s. For each step of a path, we recorded the current location, goal location, response accuracy ($RA$), and response time ($RT$) for each subject.

The *RA* was set as 1 if the optimal option was selected and set as 0 if otherwise.

## MRI data acquisition

All imaging data were collected on a Siemens Prisma$^{fit}$ 3 T MRI scanner with a 64-channel receive phased-array head/neck coil. Tight but comfortable foam padding was used to reduce head motion, and earplugs were provided to reduce acoustic noise. The stimuli were presented using a projector-mirror-screen system. The subjects viewed the screen via a mirror mounted on the head coil. Behavioral responses were recorded using a 4-button bimanual button-box.

The fMRI data were obtained using a single-shot simultaneous multi-slice (SMS) or multi-band (MB) gradient-echo EPI sequence with the following parameters: repetition time (TR) = 1500 ms, echo time (TE) = 31.0 ms, flip angle = 70°, slice acceleration factor = 3 without GRAPPA, field of view (FOV) = 211 × 211 mm², data matrix = 88 × 88, slice thickness = 2.4 mm without inter-slice gap, voxel size = (2.4 mm)³, anterior-to-posterior phase encoding direction (A»P), bandwidth = 2186 Hz/px, and 60 inter-leaved transversal slices (multi-slice mode = interleaved, and series = interleaved) covering the whole brain. To correct for susceptibility-induced geometric distortions and BOLD signal loss in the acquired functional images, we also acquired a field map of the whole brain by using a double-echo gradient-echo sequence with the following parameters: TR = 620 ms, TE1/TE2 = 4.92 ms/7.38 ms, flip angle = 60°, FOV = 211 × 211 mm², voxel size = 2.4 × 2.4 × 2.4 mm³, and 60 transverse slices. In addition, high-resolution brain structural images were acquired using a T1-weighted 3D MP-RAGE sequence with the following parameters: TR = 1800 ms, TE = 2.07 ms, flip angle = 9°, slice thickness = 0.8 mm, FOV = 256 × 256 mm², data matrix = 320 × 320, voxel size = (0.8 mm)³, and 208 sagittal slices covering the whole brain.

For each subject, the MRI scan started with a short localizer scan, followed by a resting-state fMRI (rs-fMRI) scan, a field-map, five task-fMRI runs consisting of each run for a navigation task in a specific abstract space, a T1-weighted 3D MP-RAGE scan, a T2-weighted 3D SPACE high resolution brain structural images, and a high angular resolution diffusion-weighted imaging (HARDI) scan (Fig. 1c). All of the imaging data were acquired in the same session in less than 110 min. The rs-fMRI and HARDI data were not analyzed in the current paper.

## Pre-processing of the task-fMRI data

The fMRI data were preprocessed using fMRIPrep 21.0.0[31], a standardized and efficient fMRI preprocessing pipeline based on Nipype 1.6.1[32]. For each subject, we obtained five fMRI datasets with each dataset corresponding to a navigation task in an abstract space. For a given subject, each fMRI dataset was preprocessed using the following steps. We (1) generated a reference volume and its skull-stripped brain using a custom method of fMRIPrep; (2) estimated the head-motion parameters with respect to the reference volume, including the transformation matrices, and six corresponding rotation and translation parameters, using FSL/mcflirt (version 6.0.5.1:57b01774)[33]. (In the statistical analysis, these six head-motion parameters were set as covariates to account for the residual effects of the subjects' movements.); (3) registered the magnetic field coefficients estimated from the field-map to the functional reference volume with rigid-registration; (4) corrected the slice-time of the functional images to 50% of the slice acquisition range (0.702 s, 0−1.41 s) with 3dTshift from AFNI[34]; (5) co-registered the functional reference volume to the T1-weighted 3D images using a boundary-based registration (BBR from FreeSurfer)[35] with six degrees of freedom; (6) calculated the framewise displacement (FD) and extracted the signals within the cerebrospinal fluid (CSF), white matter (WM), and gray matter (GM). (To reduce the impact of excessive head motion on the results, we set the threshold to exclude the data with an average FD > 0.25. No data was excluded according to this threshold.); (7) resampled the functional data into (2.0 mm)³ in the MNI standard space; (8) used a high pass filter with a cutoff of 1/100 Hz to remove low-frequency drifts. The images

were smoothed with an isotropic Gaussian kernel of 6 mm full-width half-maximum (FWHM) for the univariate analysis of the fMRI data. The functional images were not smoothed for the representational similarity analysis (RSA) to retain the multi-voxel pattern[36,37].

## Analyzing behavioral data in the task-fMRI scan

We sorted the subjects' *RA* and *RT*, which were recorded during the task-fMRI scanning, according to the paths in each abstract space. To study whether the subjects' behavioral performance was improved as the navigation progressed, we classified the paths into three phases: early learning (consisting of the first three paths in each space), mid-learning (consisting of the middle four paths in each space), and late learning (consisting of the last three paths in each space) (Fig. 3a).

A linear mixed-effect model (LMM) was used to capture each subject's response accuracy ($RA_{path} = \sum_{i=1}^{N_{step}} (RA_i)/N_{step}$, $i \in (1, 2, ..., N_{step})$, where $N_{step}$ represents the number of steps in the path) and response time ($RT_{path} = \sum_{i=1}^{N_{step}} (RT_i)/N_{step}$) for each path during the navigation task. We assessed whether the subjects' behavioral performance was improved as they progressed through the navigation in the multidimensional abstract spaces. LMMs are an extension of simple linear regression models that consider both fixed and random effects and are suitable for designs that include multiple observations on each subject[38]. The subsequent LMM analyses were achieved by lmerTest (version 3.1-3), an R package designed for tests in LMM[39]. The LMM was given by

$$\mathbf{Y} = \mathbf{X}_{LMM}\beta + \mathbf{Q}\gamma + \varepsilon \qquad (1)$$

where **Y**, $\mathbf{X}_{LMM}$, and **Q** represent the dependent variable, fixed effect factor, and the random effect factor, respectively. The $\beta$ and $\gamma$ represent the fixed effect and the random effect, respectively. The $\varepsilon$ represents the random error.

The first LMM analysis (LMM1) tested whether the subjects had a better performance (defined as higher $RA_{path}$ or shorter $RT_{path}$) in the late learning than in early learning phases. $RA_{path}$ and $RT_{path}$ were separately set as the dependent variable (**Y**). The early and late learning phases were set as the fixed effect factor ($\mathbf{X}_{LMM}$). The subject identity and the spatial dimension were combined and were set as the random effect factor (**Q**). In this way, we could compare whether the subjects showed significantly different behavioral performance between the early and late learning phases. The learning effect exists if a better performance was achieved in the late learning than in the early learning phase.

The second LMM analysis (LMM2) tested whether the acquired knowledge about the abstract space can be transferred into another abstract space with the same structure. The settings of the LMM2 were the same as those of the LMM1, except that the fixed effect factor ($X_{LMM}$) was the space index (Set 1 or Set 2). If the transfer learning effect existed, the subjects should respond more quickly and more accurately in the Set 2 than the Set 1.

The $RA_{step}$ or $RT_{step}$ in each path were combined into a $N_{step} \times 2$ matrix **B**.

$$\mathbf{B} = \begin{bmatrix} RA_1 & RT_1 \\ \vdots & \vdots \\ RA_{N_{step}} & RT_{N_{step}} \end{bmatrix}$$

We developed a DNN prediction model to estimate the subjects' learning level of the abstract spatial structure based on the **B** matrix. DNN is a powerful machine learning algorithm that can capture subtle characteristics in the input data by stacking layers of neural networks[40]. We first used the paths of the early learning and late learning phases to train the DNN, enabling it to learn and recognize the distinctive features of each phase. Subsequently, we used the trained DNN to estimate the probability of each path belonging to either the early or late learning phase (Eq. 3), which was defined as the learning level of the abstract space.

The DNN procedure included the following steps for each subject. First, the **B** matrix of the paths in the same dimensional spaces were grouped together and sorted according to the time sequence ([$S_{1P}$, $S_{1C}$], [$S_{2C}$, $S_{2P}$], and [$S_{3P}$], Fig. 3a). The phase of the path (early or late learning) was used as the label of the sorted data. Second, using a DNN algorithm[40], we extracted the **B** matrix patterns of the early and late learning phases. The DNN algorithm was achieved by TensorFlow (version 2.8)[41], an open-source software library for machine intelligence. Figure 3b shows that the DNN consisting of four dense layers, an input layer, two hidden layers, and an output layer. The input data (**X**$_{DNN}$) was the **B** matrix of each path in a task. One-hot encoding was used to quantify the label (**y**) to a binary $1 \times 2$ vector. The behavioral data and labels were separated into a training set and a test set. Specifically, in each dimensionality, two paths were chosen randomly as the testing set, and the other labeled paths were defined as the training set (10 paths in the 1D and 2D spaces, and 4 paths in the 3D space). The two hidden layers contained 64 and 32 units. The value of each unit in the DNN was a linear combination of the units in the previous layer, with the ReLU (rectified linear unit)[42] as the activation function for the hidden layers. The DNN was given by

$$\mathbf{X}_{i+1} = ReLU(\mathbf{w}_i^T \mathbf{X}_i + b) \tag{2}$$

where $X_i$, $w_i$, and $\boldsymbol{b}$ represent the unit matrix, weight vector, and bias respectively, of the $i^{th}$ layer of the network. The output layer contained two units, corresponding to the probabilities of the path belonging to the early and late learning phases, respectively. A predicted label (**y**$_{pred}$) was calculated from the unit values of the previous layer, with softmax[43,44] as the activation function, by using the following equation.

$$\mathbf{y}_{pred} = Softmax(\mathbf{w}_i \mathbf{X}_i + b) \tag{3}$$

In the DNN training, the weights (**w**) and biases (*b*) were initialized with random numbers in the first iteration. A predicted $1 \times 2$ vector (**y**$_{pred}$) was calculated from the input data of the training set (**X**$_{train}$) and the initialized weights and biases (Eqs. 2 and 3). The **y**$_{pred}$ was compared with the true label (**y**$_{train}$) to calculate a loss (**l**$_{train}$) using a categorical cross-entropy as the loss function[43,44]. In the next iteration, the weights and biases of the first layer were updated on the basis of the loss using the Adam (Adaptive Moment Estimation) optimizer[45]. The Adam is an algorithm that can be used to update network weights iteratively based on training data. We set the learning rate to 0.001 in the optimization. After each iteration, the values of the testing set (**X**$_{test}$) were input into the DNN. Using the weights and biases obtained from the iteration, we calculated a predicted label. The predicted label was compared with the true label (**y**$_{test}$) to calculate the loss (**l**$_{test}$)[43,44]. The model underwent 100 iterations (epochs), and the parameters were saved when the loss function reached its minimum. In each iteration, the model's prediction accuracy was computed as the ratio of correctly predicted paths to the total paths in the testing set. The prediction accuracy of the trained model was calculated by averaging the accuracy values across the 100 iterations. Finally, the **B** matrix of each path was input into the trained DNN to obtain the predicted learning level (Fig. 3b).

Figure 3c shows that a *k*-means algorithm was used to categorize the paths into the exploration and exploitation stages based on the learning level predicted by the trained DNN. The *k*-means algorithm is an unsupervised machine learning algorithm, which groups the input data into *k* clusters based on their similarities[46,47]. For each subject in each dimensional space, the predicted learning level of the paths were input to the *k*-means algorithm. An optimized *k*-means algorithm[46] was used to categorize the paths into two clusters, corresponding to the exploration and exploitation stages. The paths corresponding to the cluster with a higher probability to be early learning paths were categorized into the exploration stage, and the other cluster was defined as the exploitation stage (Fig. 3c). In this way, we categorized the paths into the exploration and exploitation stages for each subject in the 1D, 2D, and 3D spaces, separately.

After defining the exploration and exploitation stages, we tested whether the subjects' $RA_{path}$ and $RT_{path}$ were significantly different between the two stages by using a LMM analysis (LMM3). In LMM3, the exploration and exploitation stages were set as the fixed effect factor (**X**$_{LMM}$), $RT_{path}$ and $RA_{path}$ were separately set as the dependent variable (**Y**), and the combination of the subject identity and the spatial dimension was set as the random effect factor (**Q**) in the model (Eq. 1). The categorization of the two stages was considered reasonable if the exploitation stage showed a higher $RA_{path}$ or a shorter $RT_{path}$ than the exploration stage. Otherwise, if neither of these conditions was met, the categorization of the two stages would be considered as unreasonable.

### Univariate analyses of the fMRI data
The GLM analyses were carried out with FSL (version 6.0.5.1:57b01774). We set up GLM1 to examine the differences in brain activity between the exploration and exploitation stages. The navigation task included three events, which were navigation paths, feedback, and fixation periods between the paths and between trials (Fig. 2). In the subject-level GLM1 analysis, we included nine regressors for each of the five task-fMRI datasets, 1 for the paths of exploration stage, 1 for the paths of exploitation stage, 1 for the feedback, and 6 nuisance regressors for the head motion parameters. The fixation periods were treated as the baseline. The regressors for the exploration paths, exploitation paths, and feedback were convolved with the double-gamma hemodynamic response function (HRF). We contrasted the estimated parameter for the exploration paths with the baseline to obtain a COPE map (contrast of parameter estimates) for the exploration paths. Similarly, we obtained a COPE map for the exploitation path by contrasting the estimated parameter for the exploitation path with the baseline. Next, we applied a fixed effects model to compare the COPE maps between the exploration and exploitation paths across the five task-fMRI scans for each subject (exploration – baseline vs. exploitation – baseline). Finally, in the group-level GLM1 analysis, we used a random effects model to determine the brain regions with significant activation differences between the exploration and exploitation stages across all the subjects, taking the subjects' gender and age as covariates.

GLM2 was conducted to examine the association between the brain activation and the learning level predicted by the DNN. In the subject-level analysis, we included 17 regressors for each of the five task-fMRI datasets, including 10 regressors for the ten different paths, 1 for the feedback, and 6 nuisance regressors for head motion parameters. The estimated parameter for each path was contrasted to the baseline to obtain a COPE map for the path. Next, we loaded the COPE maps of the paths, set learning level as a regressor, and applied a fixed effects model to estimate the brain activation associated with the learning level. Finally, in the group-level analysis, we used a random effects model to identify brain regions with significant activation associative with the learning level. In addition, we tested the effect of spatial structure on learning by comparing brain activation associated with the learning level across the three different dimensionalities.

GLM3 was set up to detect the brain activation associative with the response accuracy of each step (*RA*). In the subject-level GLM3 analysis, we set 9 regressors for each of the five task-fMRI datasets, including 1 regressor for the navigation, 1 for the feedback, 1 for *RA* on the navigation period, and 6 nuisance regressors for head motion parameters. Next, we averaged the effect of *RA* on brain activation across the five datasets using a fixed effects model. Finally, in the group-level analysis, we used a random effects model to identify the brain region with significant activation associative with *RA*.

### Representational similarity analysis (RSA) on the fMRI data
A voxel-wise whole-brain searchlight RSA was performed separately for each stage to test whether the brain regions encoded a more accurate representation of destinations in the exploitation stage than the exploration stage. The RSA involved comparing the theoretical representational dissimilarity matrix (RDM) with the neural RDM for each brain region. First, for each subject in both stages, we constructed a theoretical RDM based on the Euclidean distance between the destinations of different paths (Fig. 4c).

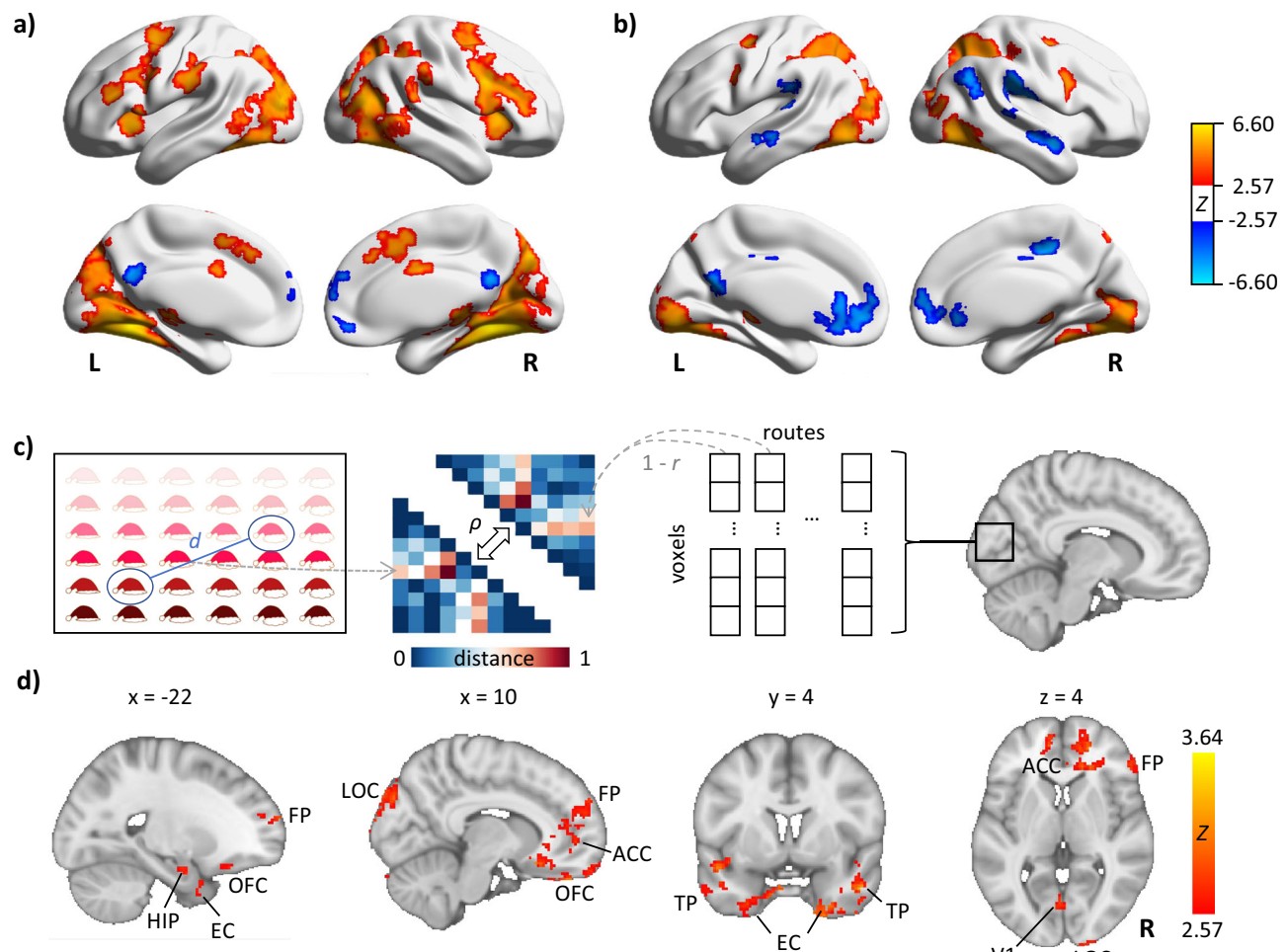

**Fig. 4 | Brain regions associated with learning in abstract spaces. a** Brain regions showing significant differences in activation between the exploration and exploitation stages. Warm (cold) colors stand for the contrast of exploration > exploitation (exploration < exploitation). **b** Brain regions with the activation associated with response accuracy. Warm (cold) colors stand for a positive (negative) association. The color bar represents the *Z*-values. **c** Schematic for the representational similarity analysis (RSA). The lower triangular matrix depicts a theoretical representational dissimilarity matrix (RDM) where each element was the Euclidean distance (*d*) between goals of different paths. The upper triangular matrix depicts a neural RDM, with each element was the dissimilarity (1 - *r*) between two paths, where *r* is the

Pearson's correlation coefficient. The neural RDM was constructed by calculating the dissimilarity of voxel-wise parametric estimations within brain regions between paths. The Spearman's rank correlation coefficient (*ρ*) was then computed between the theoretical RDM and the neural RDM. **d** Brain regions identified through voxel-wise searchlight RSA in the whole-brain. Significant regions indicate improved representation on destinations after learning. The color bar represents the *Z*-values. The underlying statistical maps are available at https://identifiers.org/neurovault.collection:16948. Abbreviations: HIP hippocampus, EC entorhinal cortex, OFC orbitofrontal cortex, FP frontal pole, ACC anterior cingulate cortex, LOC lateral occipital cortex, TP temporal pole, V1 primary visual cortex.

Second, for a given brain region, we calculated a neural RDM for each subject in both stages. To achieve this, we conducted the subject-level GLM2 analysis on the unsmoothed preprocessed fMRI data, resulting in a parameter estimation (PE) map for each path. We then used NeuroRA[48], a Python toolbox designed for performing RSA, to calculate the dissimilarity (1 - *r*) between the PE maps of each path, obtaining the neural RDM. Third, we computed Spearman's rank correlation (*ρ*) between the neural RDM and the theoretical RDM for each subject. The searchlight size was set to a 3-voxel cube, with a stride of 1 voxel along each axis (*x*, *y*, and *z*). For each stage, we obtained a whole brain *ρ*-map representing the correlation between the neural RDM and the theoretical RDM for each subject through searchlight RSA. After applying Fisher's transformation to standardize the *ρ*-maps, we compared the *ρ*-maps between the two stages to identify brain regions that showed a more accurate representation of destinations in the exploitation stage than the exploration stage.

**Statistics and reproducibility**

For the behavioral data, a paired-sample *t*-test was used to estimate the significance of fixed-effect in the LMM. A one-sample *t* test was used to test

whether the prediction accuracy of the DNN model was significantly higher than the random level, which was set at 0.25 (=0.5/2). The reason was that, at the random level, the DNN model has a probability of 0.5 to select the correct label, and the DNN model needed to judge the 2 paths of the testing set in each iteration (detail described in *Learning level estimated by the deep neural network algorithm*). A repeated measure analysis of variance (ANOVA) was used to test the difference in the prediction accuracy of the trained DNN across the three dimensionalities. A paired-sample *t* test was used to compare the path numbers of the exploration and exploitation stages categorized by the *k*-means algorithm. The significant level was set at *p* < 0.05.

For the GLM analysis of the fMRI data, we performed a paired-sample *t* test to test the differences in brain activations between the exploration and exploitation stages, and a one-sample *t* test to test the effect of the learning level and *RA* on the brain activation. A Gaussian random field (GRF) correction was used to control for multiple comparisons. The significance level was set at *p* < 0.05, with the cluster forming threshold at *p* < 0.001. A repeated measures ANOVA was used to test whether the brain activation modulated by the learning level was different among the three

**Table 1 | Analysis of behavioral data in the scanner using linear mixed effect models (LMM), with a sample size of 25 subjects**

| Response | β | Standard error | *t* value | *p* value |
|---|---|---|---|---|
| *Response accuracy* | | | | |
| LMM1 (early vs. late learning) | −0.13*** | 0.03 | −3.89 | <0.001 |
| LMM2 (Set 1 vs. Set 2) | −0.14*** | 0.03 | −4.86 | <0.001 |
| LMM3 (exploration vs. exploitation) | −0.15*** | 0.03 | −5.63 | <0.001 |
| *Response time* | | | | |
| LMM1 (early vs. late learning) | 0.22 | 0.13 | 1.76 | 0.079 |
| LMM2 (Set 1 vs. Set 2) | −0.03 | 0.09 | −0.37 | 0.711 |
| LMM3 (exploration vs. exploitation) | 0.17* | 0.08 | 2.06 | 0.040 |

LMM1 tested whether the subjects had a higher $RA_{path}$ or a shorter $RT_{path}$ in the late learning than in early learning phases. LMM2 tested whether the acquired knowledge about the abstract space can be transferred into another abstract space with the same dimensionality. LMM3 tested whether the subjects' $RA_{path}$ and $RT_{path}$ were significantly different between the exploration and exploitation stages. The general form of the LMM was given by $\mathbf{Y} = \mathbf{X}_{LMM}\beta + \mathbf{Q}\gamma + \varepsilon$ (Eq. 1), where $\mathbf{Y}$, $\mathbf{X}_{LMM}$, and $\mathbf{Q}$ represent the dependent variable, fixed effect factor, and the random effect factor, respectively; and $\beta$, $\gamma$, and $\varepsilon$ represent the fixed effect, random effect, and random error, respectively. In LMM1, the dependent variable ($\mathbf{Y}$) corresponded to the response accuracy ($RA_{path}$) or response time ($RT_{path}$), the fixed effect factor ($\mathbf{X}_{LMM}$) corresponded to the early and late learning phases, and the random effect factor ($\mathbf{Q}$) corresponded to the subject identity and spatial dimension. For LMM2 and LMM3, the settings were similar to those of LMM1, except for the fixed effect factor, which was the space index (Set 1 or Set 2) for LMM2 and the stage of the path (exploration or exploitation) for LMM3. *$p$ < 0.05; **$p$ < 0.01; ***$p$ < 0.001.

dimensionalities. For the searchlight RSA, a paired-sample *t* test was used to compare the destination representation between exploration and exploitation stages. Threshold-free cluster enhancement (TFCE) was used to identify the statistically significant cluster size (10,000 permutations) using the family-wise error (FWE) method for multiple comparisons correction.

## Reporting summary
Further information on research design is available in the Nature Portfolio Reporting Summary linked to this article.

## Results
### Behavioral performance in the scanner
Overall, the subjects' performance improved as the task progressed in the abstract spaces of each dimensionality (Fig. 2b and Table S3 in Supplementary Information). From the LMM1 analysis, we found significantly lower $RA_{path}$ in the early learning than late learning phase ($t = −3.89$, $p < 0.001$). The LMM2 analysis showed significantly lower $RA_{path}$ in Set 1 than in Set 2 ($t = −4.86$, $p < 0.001$). No significant difference in $RT_{path}$ was found either between the early and late learning phases from LMM1 ($t = 1.76$, $p = 0.079$) or between Set 1 and Set 2 from LMM2 ($t = −0.37$, $p = 0.711$). The detailed information is listed in Table 1.

The paths were divided into the exploration and exploitation stages using a DNN model and a *k*-means algorithm. The DNN model showed a significantly higher prediction accuracy than the chance level ($t = 41.74$, $p < 0.001$, Fig. 3d), and there were no significant differences in the prediction accuracy across the three dimensionalities ($F_{(2, 48)} = 8.68$, $p = 0.134$). Based on the behavioral performance characteristics extracted by the DNN, the *k*-means algorithm assigned 50.32% of the paths to the exploration stage ($24.84 \pm 3.53$), while the remaining 49.68% were assigned to the exploitation stage ($25.16 \pm 3.53$) (Fig. 3e and Table S2, Supplementary Information). At the group-level, there were no significant difference in the number of paths between the two stages for any of the three dimensionalities (1D: $t = −0.92$, $p = 0.365$; 2D: $t = 0.71$, $p = 0.485$; 3D: $t = 0.12$, $p = 0.903$). From LMM3, we found that $RA_{path}$ was significantly lower in the exploration stage than the

exploitation stage ($t = −5.63$, $p < 0.001$, Table 1). Additionally, the $RT_{path}$ was significantly larger in the exploration stage than the exploitation stage ($t = 2.06$, $p = 0.040$). These results indicated the reasonability of separating the paths into the exploration and exploitation stages.

### Univariate analysis of the fMRI data
From the GLM1 analysis, we observed 13 clusters that showed significantly stronger activation in the exploration stage than the exploitation stage (Fig. 4a). The largest cluster was primarily located in the right HIP and extended to the left HIP (peak MNI coordinates $(x, y, z) = (28, −66, −16)$). The other clusters were located in the bilateral inferior frontal gyrus (IFG) (left: −24, 2, 60; right: 42, 12, 28), bilateral insula (left: −36, 16, 0; right: 34, 20, 4), bilateral medial frontal gyrus (MeFG) (0, 20, 42), bilateral inferior parietal lobule (IPL) (left: −58, −24, 32; right: 62, −32, 38), left middle frontal gyrus (MiFG) (−42, 30, 18), right ventral anterior cingulate cortex (ACC) (4, −2, 30), right thalamus (32, −14, −6), and bilateral cerebellum (left: −16, -38, −48; right: 16, −72, −46). Furthermore, we identified two clusters that showed significantly weaker activation in the exploration stage than the exploitation stage. These clusters were located in the right posterior cingulate cortex (PCC) (2, −48, 22) and in the right superior frontal gyrus (SFG) (2, 56, 26). The detailed information about these clusters is listed in Supplementary Data 2.

From the GLM2 analysis, we identified brain regions with activity significantly associated with the learning level, which corroborated the findings from the GLM1 analysis (Fig. S5, Supplementary Information). We detected 11 clusters showing positive association with the learning level. These clusters encompassed the right HIP (28, −46, −8), bilateral MiFG (left: −38, 30, 12; right: 24, 6, 46), bilateral insula (left: −26, 6, 0; right: 30, 20, −8), left SFG (−18, −6, 50), left MeFG (−6, 12, 46), left postcentral gyrus (PoCG) (−60, −28, 40), right IPL (56, −36, 18), and bilateral cerebellum (left: −16, −38, −48; right: 18, −76, −48). Additionally, we identified two clusters in the left MeFG (−2, 58, 10) and right PCC (4, −50, 24), with negative association with the learning level. The detailed information about these clusters is listed in Supplementary Data 3. No brain region showed significantly different learning-associative activation across the three dimensionalities.

From the GLM3 analysis, we observed seven clusters showing positive association with $RA$ (Fig. 4b). These clusters were located in the right lingual gyrus (20, −84, 4), bilateral MiFG (left: −38, −6, 50; right: 28, −2, 48), bilateral IFG (left: −46, 4, 32; right: 48, 10, 30), right thalamus (24, −26, −2), and left HIP (−20, −32, −2). In addition, we detected nine clusters showing negative association with $RA$. These clusters encompassed the left ACC (4, 56, 4), bilateral IPL (left: 56, −28, 26; right: −66, −28, 22), right PCC (12, −30, 44), left retrosplenial cortex (RSC) (−12, −58, 14), bilateral middle temporal gyrus (MTG) (left: −58, −20, −10; right: 54, −2, −14), right superior temporal gyrus (STG) (50, −62, 30), and right parahippocampal gyrus (PHG) (26, −48, 12). The detailed information about these clusters is listed in Supplementary Data 4.

### RSA on the fMRI data
Using whole-brain searchlight RSA, we identified 14 clusters that showed a significantly more accurate representation of destinations in the exploitation stage than the exploration stage (Fig. 4d). These clusters were located in the bilateral entorhinal cortex (left: −16, −2, −22; right: 34, −2, −34), bilateral cuneus (left:−4, −68, 8; right: 8, −86, 34), left inferior temporal gyrus (ITG) (−54, −20, −36), right ACC (22, 34, 6), left frontal pole (−16, 64, 18), bilateral SFG (left: −8, 54, 40; right: 22, 38, 24), bilateral OFC (left: −4, 70, −8; right: 6, 36, −30), left IFG (-30, 32, -22), and right cerebellum (42, −82, −30). The detailed information about these clusters is listed in Supplementary Data 5.

## Discussion
This study analyzed the subjects' behavioral performance and revealed different brain activation patterns during the exploration and exploitation stages in the abstract spaces (Fig. 1). The subjects performed a navigation

task in five different abstract spaces during fMRI scanning (Fig. 2). We conducted two LMM analyses to capture the subject's behavioral performance and found that the subjects achieved higher accuracy as the navigation progressed (Table 1). To reveal the neural mechanisms supporting this improvement in response accuracy, we separated the navigation paths into exploration and exploitation stages by combining the DNN and *k*-means algorithms (Fig. 3). Using a GLM analysis, we compared the difference in the brain activity between the two stages and found that (1) the bilateral HIP, PFC, and visual cortex had significantly stronger activation in the exploration than in exploitation and (2) the bilateral medial OFC and the RSC had significantly stronger activation in the exploitation than in the exploration (Fig. 4a, GLM1). These differences were also related to the learning level predicted by the DNN (Fig. S5, GLM2), but not related to the dimensionalities. Activation in the bilateral HIP and visual cortex were positively correlated to the response accuracy, while activation in the bilateral OFC and RSC was negatively correlated to the response accuracy (Fig. 4b, GLM3). Using the RSA method, we found that the bilateral OFC, bilateral entorhinal cortex, and left HIP form more accurate representation to the navigational destinations in the exploitation than in the exploration stage (Fig. 4d).

Regarding the learning in the abstract spaces, from LMM1, we detected a higher path accuracy in the exploitation than the exploration (Table 1 and Fig. S4), indicating that the subjects became familiar with the abstract space as the task proceeded. This result indicates that subjects may refine their maps of an abstract space as they explore from a first-person perspective. This is consistent with previous observations in both physical and abstract spaces[23,24,49,50].

We detected a transfer learning effect across the navigation task in the abstract spaces (Table 1 and Fig. S4). From LMM2, a higher path accuracy was detected in Set 1 ($S_{1P}$ and $S_{2C}$) than in Set 2 ($S_{1C}$ and $S_{2P}$). This result may indicate that the acquired information can be transferred from one space to another space with the same structure, even though the spaces look entirely different. This result is consistent with previous ones that showed that an internal representation of a task structure (e.g., a solution of a problem) could be transferred to a new situation when the structure was actually the same even though the situations looked different[51,52].

The navigation paths were separated into exploration and exploitation stages using a combination of the DNN prediction model and the *k*-means algorithm. The results showed an improvement in both $RA_{path}$ and $RT_{path}$ in the exploitation than in the exploration stage (Table 1). This separation outcome indicated the inseparability of exploration and exploitation in a sequential manner, where an exploitation path may follow an exploration path. These results align with the trade-off between exploration and exploitation in reinforcement learning theory[53]. Exploration enables individuals to gather more information about the environment and ultimately improve long-term performance. However, pure exploration strategies may lead to prolonged uncertainty and inefficiency. Conversely, exploitation uses existing knowledge and experience to efficiently select the currently optimal choice for maximizing immediate rewards. However, relying solely on exploitation may cause individuals to miss out on potentially better alternatives[54]. Our results show that the exploration and exploitation are intertwined during the learning process of abstract spaces.

Regarding the brain activation related to the learning, from GLM1 analysis, we observed distinct brain regions engaged in the exploration and exploitation stages (Fig. 4a). In the exploration, the HIP, lateral PFC, insula, thalami, IPL, and visual cortex showed significantly stronger activation than in the exploitation. These regions likely form a network for mapping abstract spaces during the exploration. Previous studies highlighted the complementary roles of the PFC and HIP in learning and inference[12,18,19,55] and constructing cognitive maps[15,24,56]. Additionally, the insula, visual cortex, thalamus, and IPL were implicated in various learning processes[57–60] and coupled with the PFC and HIP in explicit memory[61–63].

In the exploitation, we observed stronger activation in the bilateral mPFC, OFC, and RSC than in the exploration (Fig. 4a). In the exploitation stage, individuals primarily need to compare sensory inputs to the memory retrieved based on the formed map of abstract spaces. The mPFC is a higher-level brain region associated with outcome evaluation, goal planning, action execution, and event prediction[64,65]. The OFC is involved in inference and formation of cognitive maps in abstract spaces[25,29], while the RSC plays a major role in anchoring the cognitive map in mind to the actual environment and facilitating perspective shifts[66,67]. Furthermore, we found that the brain regions significantly activated in the exploration and exploitation stages, including the HIP, OFC, and RSC, were associative with the learning level predicted by the DNN (GLM2, Fig. S5). This result further supports their involvement in forming cognitive map in abstract spaces.

From GLM3 analysis, we also observed that activation in the HIP, lateral PFC, and visual cortex positively correlated with response accuracy, while activation in the OFC and RSC showed a negative correlation with response accuracy (Fig. 4b). The HIP has been implicated in representing distance to goals in both physical and abstract spaces[10,49,68,69]. The visual cortex, along with the PHG, is involved in estimating distance[70]. The OFC is necessary for forming appropriate behavior-outcome associations[71].

The HIP and OFC have been proposed as core regions for encoding cognitive maps[29]. From three GLM analyses (Fig. 4), we found different activation patterns of the HIP and OFC. The HIP exhibited greater engagement in the exploration stage or a lower learning level, and its activation was positively associated with behavioral performance. The OFC showed greater engagement in the exploitation stage or a higher learning level, and its activation was negatively associated with behavioral performance. These results indicated that these two regions may adopt different roles in forming and representing a cognitive map. Previous studies suggested that the HIP and OFC are involved in different aspects of a cognitive map[29,72,73]. Our results indicated that the HIP may be responsible for updating the map of spaces and the OFC for utilizing the map to make inferences. In our recent study[72], we also found that the HIP was involved in collecting and updating information, whereas the OFC was associated with the relationships between agents in the environment.

During the goal-directed navigation in abstract spaces, we found that several brain regions, including the HIP, entorhinal cortex, OFC, and LOC, showed more accurate destination representations in the exploitation stage than the exploration stage (Fig. 4d). When subjects navigated to destinations of nearby locations, these brain regions displayed more similar activation patterns in the exploitation stage. The HIP and entorhinal cortex are well-established central regions for cognitive map-based navigation[68,74,75]. The OFC was also found representing the future goals during spatial navigation[76], and cooperating with the HIP in forming cognitive maps[73,77]. The LOC was suggested as the region for representing object identity, encoding perceptual information of visual stimuli, and coding object interaction[78–80]. Though the LOC was not a typical brain region of cognitive map, previous studies also found that the LOC was essential in processing boundaries and navigational affordances[81–83]. These regions may be involved in encoding and representing spatial information, including the locations of destinations.

The following limitations of the current study should be addressed. First, the abstract spaces used in the current study were constructed artificially, so they may not fully reflect the complex concepts in the real world. We took measures to encourage the subjects integrating different dimensional information during making decisions. Nevertheless, we cannot rule out the possibility that the subjects may consider the information of different dimensions separately. Second, the BOLD signal in the brain regions (HIP, EC, and OFC) that we primarily focused on is sensitive to the fMRI settings, such as hardware, multi-band acceleration factors of the gradient-echo EPI sequence, BOLD sensitivity related to slice orientation and susceptibility effect, and the resolution of the fMRI[84–89].

The current study revealed the behavioral and neural patterns during learning in abstract spaces. We found that the subjects showed better behavioral performance as they progressed through the navigation process in multidimensional abstract spaces, indicating a potential refinement of the internal representation of the abstract spaces. A brain network including the hippocampus, lateral prefrontal cortex, and visual cortex showed stronger

activation in exploration than exploitation, and the activation of these regions were positively associative with response accuracy. Another brain network including the medial prefrontal cortex, orbitofrontal cortex, and retrosplenial cortex showed stronger activation in exploitation than exploration, and the activation of these regions were negatively associative with response accuracy. The hippocampus, entorhinal cortex, and orbitofrontal cortex showed more accurate representation to the navigational destinations with the learning in the abstract spaces. These findings may help understanding how our brain encode the encountered environments or problems.

## Data availability
The data that support the findings of this study are available from the corresponding author upon reasonable request. Source data underlying Figs. 2 and 3 are supplied in Supplementary Data 1. Statistical maps underlying Fig. 4 are supplied at https://identifiers.org/neurovault.collection:16948.

## Code availability
The navigation task programs and data analysis codes are available at https://github.com/YidanQiu/AbsNvg.git[90].

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

## Acknowledgements
This work was supported by funding from the National Natural Science Foundation of China (Grant numbers: 32371101 and 82171914), Guangdong Natural Science Foundation (2022A1515011022), and National Key Research and Development Program of China (Grant number: 2018YFC1705000). The authors appreciate Rhoda E. and Edmund F. Perozzi, PhDs, for editing the manuscript.

## Author contributions
Yidan Qiu: Conceptualization, Investigation, Methodology, Formal analysis, Writing-original draft, Writing-review and editing, Project administration. Huakang Li: Methodology, Data curation, Formal analysis, Writing-review and editing. Jiajun Liao: Methodology, Validation, Writing-review and editing. Kemeng Chen: Investigation, Writing-review and editing Xiaoyan Wu: Validation, Writing-review and editing. Bingyi Liu: Writing-review and editing. Ruiwang Huang: Conceptualization, Resources, Supervision, Project administration, Funding acquisition.

## Competing interests
The authors declare no competing interests.
