## [Peer Review File · Communications Biology]

Reviewers' comments:

Reviewer #1 (Remarks to the Author):

The authors present a study on how cognitive maps can be created and how such a space can be navigated. They have created a novel and clever task which allowed participants to transverse a concrete visual space as a cognitive map. The article reads clearly, despite the use of advanced statistical analyses. For the most part, the authors present analyses that are appropriate for their hypotheses and mostly support their claims. There are a few major comments that should be addressed to better align the authors' hypotheses, analyses and claims. Lastly, the authors present work that will be of great interest to people in many subfields (memory, decision making, spatial navigation). I believe the authors could address some more general implications of their findings to these fields.

Major comments

1. The categorization of paths seems arbitrary for me. Especially when there is a difference in the SP1 and SP2 scenarios. Do paths 5 and 6 really have as big of a difference in learning as paths 1 and 10 in SP1? Would it not be safer to just use paths 1-3 and 8-10 in both SP1 and SP2. Then you have behavior that is closer to the two ends of "learning" and an equal sample between the SP1 and SP2 scenarios. Not sure if this would be a better solution or not. If the authors want to stick with their current definitions of path EEP and LEP labels, perhaps they could provide some sources that would back up a median split for learning effects or anything else to justify their assumption.
2. I do not believe that your analyses address your hypothesis, "...if the HIP, EC, and OFC show different activation strengths and representation patterns between two learning stages, they are involved in learning the structure of abstract spaces". While you do show results in support of the first part of the hypothesis, there is no analysis linking the fMRI data directly to any kind of learning parameter. I believe the authors show this indirectly, by showing that the earlier and later learning stages are related to better accuracy and reaction times from the LMM models. But there is no direct link of behavior to neural data. I believe this should be easy to do with either the univariate or the multivariate analyses.
3. In the discussion, on page 36-37, the authors make a claim about how the anterior and posterior hippocampus show different levels of representation of the two stages, and made a claim that this may indicated specialization. I believe this claim could be support by the data, but the authors did not complete that analysis. A simple analysis to test whether the anterior and posterior hippocampus have significantly different levels of representing learning stages could support their claim.

Minor comments

1. There should be more justification for why the RSA had to be done. Why is showing a pattern analysis difference unique from showing a magnitude response difference such as the univariate analysis. Perhaps the authors wanted to utilize multi-voxel pattern analysis techniques because it can be more sensitive, particularly to dimensional concepts. Further, differences in the whole-brain univariate and RSA analyses should be highlighted if the RSA did reveal something that for some theoretical reason wouldn't be revealed from the univariate analysis.

a. Chikazoe, J., Lee, D.H., Kriegeskorte, N., Anderson, A.K. (2014). Population coding of affect across stimuli, modalities and individuals. *Nature Neuroscience*, 17(8), 1114–22. <https://doi.org/10.1038/nn.3749>.

b. Popal, H., Wang, Y., Olson, I.R., 2019. A guide to representational similarity analysis for social neuroscience. *Social Cogn. Affect. Neurosci.* 14(11), 1243–1253. <https://doi.org/10.1093/scan/nsz099>.

2. How do the authors' findings map onto other forms of cognitive maps? Do these results speak to any kind of domain general ability of the medial temporal lobe system and OFC to create cognitive maps? Are these results similar to cognitive map studies showing more abstract knowledge maps? Are these results relevant to spatial navigation or understanding social networks?

Reviewer #3 (Remarks to the Author):

The authors utilized fMRI to examine how people construct abstract maps. They designed navigation tasks in 1D, 2D, and 3D abstract spaces and acquired fMRI data while subjects explored an abstract space. The authors first employed a deep neural network to capture the behavioral path pattern in the navigation and used a k-means algorithm to separate the task into pre- and post-learning stages. Then they compared the brain activations and representations between the pre- and post-learning stages. They found significantly different activations between the two stages that included the bilateral hippocampus, lateral and medial prefrontal cortex, and precuneus. The hippocampus also showed different representations of the paths of the two stages.

Overall, I am not convinced that the claims made are supported by the data. Throughout, I found the data presentation confusing and unclear.

Major issues:

First, I don't understand why DNN was used to analyze behavioral data. The authors convert the response accuracy (RA_{step}) and response time (RT_{step}) on a path into a 2-dimensional matrix and give it as input to a classifier. However, DNN itself is a black box, and this analysis still doesn't tell us which behavioral features change fundamentally, compared to fig2b.

Second, the authors just analyze the learning effect, which is present in many experimental tasks. They compared activation in early and late stages and therefore argued that this change reflects which brain regions construct cognitive maps. Obviously, this simple analysis cannot support the conclusion. The key to the construction of cognitive maps lies in how to learn the connections between different stimuli, for example, using RSA to analyze whether the neural similarities between different stimuli conform to the metric features of cognitive maps or to reveal the neural geometric structures between stimuli. Then to analyze how these neural geometric structures evolve with learning.

Third, most studies of abstract space involve the integration of different dimensions, that is, the linkages between stimuli. In this experiment, however, subjects did not seem to need to integrate different dimensions. Therefore, I do not think this is an elegant design for studying abstract space construction. The decision at each step can be seen as a simple value decision.

Response to Reviewers' comments

Manuscript Number: COMMSBIO-23-2026

Manuscript Title: Forming a cognitive map for an abstract space: the roles of the human hippocampus and orbitofrontal cortex

Reviewer #1:

The authors present a study on how cognitive maps can be created and how such a space can be navigated. They have created a novel and clever task which allowed participants to transverse a concrete visual space as a cognitive map. The article reads clearly, despite the use of advanced statistical analyses. For the most part, the authors present analyses that are appropriate for their hypotheses and mostly support their claims. There are a few major comments that should be addressed to better align the authors' hypotheses, analyses and claims. Lastly, the authors present work that will be of great interest to people in many subfields (memory, decision making, spatial navigation). I believe the authors could address some more general implications of their findings to these fields.

Response: Thank you very much for your positive comments. Following your suggestions, we have answered your questions/concerns one by one and revised the manuscript accordingly.

Q1. The categorization of paths seems arbitrary for me. Especially when there is a difference in the SP1 and SP2 scenarios. Do paths 5 and 6 really have as big of a difference in learning as paths 1 and 10 in SP1? Would it not be safer to just use paths 1-3 and 8-10 in both SP1 and SP2. Then you have behavior that is closer to the two ends of “learning” and an equal sample between the SP1 and SP2 scenarios. Not sure if this would be a better solution or not. If the authors want to stick with their current definitions of path EEP and LEP labels, perhaps they could provide some sources that would back up a median split for learning effects or anything else to justify their assumption.

Response: Thank you for your valuable suggestions. We agree with your recommendation to redefine the paths in a more consistent manner. Therefore, we have revised our categorization approach to designate paths 1-3 in each space as early learning phase (formerly referred to as “EEP”), paths 8-10 in each space as late learning phase (formerly referred to as “LEP”), and the remaining paths as mid-learning phase. This new categorization provides a clearer distinction and aligns with the desired behavior representing the extremes of “learning”. We believe that this modification enhances the validity of our study.

Accordingly, we have revised the description on the definition of the learning phases on page 14, line 13 as follows:

“To study whether the subjects’ behavioral performance was improved as the navigation progressed, we classified the paths into three phases: early learning

(consisting of the first three paths in each space), mid-learning (consisting of the middle four paths in each space), and late learning (consisting of the last three paths in each space) (Fig. 3a).”

After revising the categorization of paths, we re-analyzed the behavioral data by using the linear mixed-effect model (LMM). We are pleased to report that the revised analysis yielded results consistent with our original findings. Specifically, we observed a clear learning effect as the task progressed. Now, we have revised the description on page 24, line 13 accordingly, as follows:

“From the LMM1 analysis, we found significantly lower RA_{path} in the early learning than late learning phase ($t = -3.89, p < .001$). The LMM2 analysis showed significantly lower RA_{path} in Set 1 than in Set 2 ($t = -4.86, p < .001$). No significant difference in RT_{path} was found either between the early and late learning phases from LMM1 ($t = 1.76, p = .079$) or between Set 1 and Set 2 from LMM2 ($t = -0.37, p = .711$). The detailed information is listed in Table 1.”

Next, we employed a deep neural network (DNN) to assess the learning level exhibited by the subjects regarding the structure of the abstract space. To train the DNN, we utilized the labeled paths representing the early and late learning categories. These paths served as the basis for teaching the DNN to recognize and understand the underlying behavioral patterns and characteristics of the subject’s learning in abstract spaces. Once the DNN was trained, we utilized it to estimate the learning level, which was the probability of the path being categorized to the early or late learning category, for all paths within the spaces. The corresponding description has been revised on page 16, line 12 as follows:

“The RA_{step} or RT_{step} in each path were combined into a $N_{step} \times 2$ matrix \mathbf{B} .

$$\mathbf{B} = \begin{bmatrix} RA_1 & RT_1 \\ \vdots & \vdots \\ RA_{N_{step}} & RT_{N_{step}} \end{bmatrix}$$

We developed a DNN prediction model to estimate the subjects’ learning level of the abstract spatial structure based on the \mathbf{B} matrix. DNN is a powerful machine learning algorithm that can capture subtle characteristics in the input data by stacking layers of neural networks (Schmidhuber, 2015). We first used the paths of the early learning and late learning phases to train the DNN, enabling it to learn and recognize the distinctive features of each phase. Subsequently, we used the trained DNN to estimate the probability of each path belonging to either the early or late learning phase (Eq. 3), which was defined as the learning level of the abstract space.”

In this way, 50.32% of all paths were assigned to the exploration stage, while the remaining 49.68% of paths were assigned to the exploitation stage. As expected, the subjects exhibited better behavioral performance in the exploitation stage compared to the exploration stage. We have revised the results accordingly on page 24, line 20, as follows:

“The paths were divided into the exploration and exploitation stages using a DNN model and a k -means algorithm. The DNN model showed a significantly higher prediction accuracy than the chance level ($t = 41.74, p < .001$), and there were no significant differences in the prediction accuracy across the three dimensionalities ($F_{(2, 48)} = 8.68, p = .134$). Based on the behavioral performance characteristics extracted by the DNN, the k -means algorithm assigned 50.32% of the paths to the exploration stage (24.84 ± 3.53), while the remaining 49.68% were assigned to the exploitation stage (25.16 ± 3.53) (Fig. 3e and

Table S2, Supplementary Materials). At the group-level, there were no significant difference in the number of paths between the two stages for any of the three dimensionalities (1D: $t = -0.92, p = .365$; 2D: $t = 0.71, p = .485$; 3D: $t = 0.12, p = .903$). From LMM3, we found that RA_{path} was significantly lower in the exploration stage than the exploitation stage ($t = -5.63, p < .001$, Table 1). Additionally, the RT_{path} was significantly larger in the exploration stage than the exploitation stage ($t = 2.06, p = .040$). These results indicated the reasonability of separating the paths into the exploration and exploitation stages.”

We also reanalyzed the fMRI data using the revised categorization of the exploration and exploitation stages. The updated results can be found on page 25, line 16, and are described as follows:

“From the GLM1 analysis, we observed 13 clusters that showed significantly stronger activation in the exploration stage than the exploitation stage (Fig. 4a). The largest cluster was primarily located in the right HIP and extended to the left HIP (peak MNI coordinates $(x, y, z) = (28, -66, -16)$). The other clusters were located in the bilateral inferior frontal gyrus (IFG) (left: -24, 2, 60; right: 42, 12, 28), bilateral insula (left: -36, 16, 0; right: 34, 20, 4), bilateral medial frontal gyrus (MeFG) (0, 20, 42), bilateral inferior parietal lobule (IPL) (left: -58, -24, 32; right: 62, -32, 38), left middle frontal gyrus (MiFG) (-42, 30, 18), right ventral anterior cingulate cortex (ACC) (4, -2, 30), right thalamus (32, -14, -6), and bilateral cerebellum (left: -16, -38, -48; right: 16, -72, -46).

Furthermore, we identified two clusters that showed significantly weaker activation in the exploration stage than the exploitation stage. These clusters were located in the right posterior cingulate cortex (PCC) (2, -48, 22) and in the right superior frontal gyrus (SFG) (2, 56, 26). The detailed information about these clusters is listed in Table S4 (Supplementary Materials).”

Table 1. Behavioral data in the scanner analyzed using linear mixed effect models (LMM).

Response	β	Standard error	t -value	p -value
Response accuracy				
LMM1 (early vs. late learning)	-0.13***	0.03	-3.89	< .001
LMM2 (Set 1 vs. Set 2)	-0.14***	0.03	-4.86	< .001
LMM3 (exploration vs. exploitation)	-0.15***	0.03	-5.63	< .001
Response time				
LMM1 (early vs. late learning)	0.22	0.13	1.76	.079
LMM2 (Set 1 vs. Set 2)	-0.03	0.09	-0.37	.711
LMM3 (exploration vs. exploitation)	0.17*	0.08	2.06	.040

Notes: LMM1 tested whether the subjects had a higher RA_{path} or a shorter RT_{path} in the late learning than in early learning phases. LMM2 tested whether the acquired knowledge about the abstract space can be transferred into another abstract space with the same dimensionality. LMM3 tested whether the subjects' RA_{path} and RT_{path} were significantly different between the exploration and exploitation stages. The general form of the LMM was given by $\mathbf{Y} = \mathbf{X}_{LMM}\beta + \mathbf{Q}\gamma + \varepsilon$ (eq. 1), where \mathbf{Y} , \mathbf{X}_{LMM} , and \mathbf{Q} represent the dependent variable, fixed effect factor, and the random effect factor, respectively; and β , γ , and ε represent the fixed effect, random effect, and random error, respectively. In LMM1, the dependent variable (\mathbf{Y}) corresponded to the response accuracy (RA_{path}) or response time (RT_{path}), the fixed effect factor (\mathbf{X}_{LMM}) corresponded to the early and late learning phases, and the random effect factor (\mathbf{Q}) corresponded to the subject identity and spatial dimension. For LMM2 and LMM3, the settings were similar to those of LMM1, except for the fixed effect factor, which was the space index (Set 1 or Set 2) for LMM2 and the stage of the path (exploration or exploitation) for LMM3. *, $p < .05$; **, $p < .01$; ***, $p < .001$.

Fig. 3. Schematic and results for the separation of the navigation paths into the exploration and exploitation stages. **(a)** Definition of the early learning (green), mid-learning (white), and late learning (orange). The first three paths and the last three paths of each space were labeled as the early and late learning phases, respectively. These labeled paths were used in training the deep neural network (DNN). **(b)** Construction of the DNN prediction model. The DNN contains 2 hidden layers between the input and output layers. The input data were a N_{step} -by-2 matrix $\mathbf{B} = [RA_1, RT_1; \dots; RA_{N_{step}}, RT_{N_{step}}]$ of a path, where $RA_{N_{step}}$ and $RT_{N_{step}}$ represent response accuracy and response time of the N^{th} step, respectively. The values of the two units in the output layer indicate the probability of the path being categorized as early learning phase (denoted as $P(early)$) and being categorized as late learning phase (denoted as $P(late) = 1 - P(early)$). The \mathbf{B} matrices of all paths

were input to the trained DNN to obtain the $P(early)$. **(c)** Categorization the navigation paths into the exploration and exploitation stages. A k -means algorithm was used to categorize the paths into the exploration and exploitation stages based on the $P(early)$. **(d)** Performance of the DNN prediction model. During training, the DNN model predicted the label of the paths in the test set. The accuracy was calculated as the ratio of correctly predicted paths to the total paths in the testing set and averaged across the 100 iterations. The model accuracy value was significantly higher than the chance level ($0.25 = 0.5 / n_{test}$). **(e)** The number of paths categorized as exploitation stage by the k -means algorithm in each space. $***, p < .001$.

Fig. 4. Brain regions associated with learning in abstract spaces. **(a)** Brain regions showing significant differences in activation between the exploration and exploitation stages. Warm (cold) colors stand for the contrast of “exploration > exploitation”

(“exploration < exploitation”). **(b)** Brain regions with the activation associated with response accuracy. Warm (cold) colors stand for a positive (negative) association. The color bar represents the Z -values. **(c)** Schematic for the representational similarity analysis (RSA). The lower triangular matrix depicts a theoretical representational dissimilarity matrix (RDM) where each element was the Euclidean distance (d) between goals of different paths. The upper triangular matrix depicts a neural RDM, with each element was the dissimilarity ($1 - r$) between two paths, where r is the Pearson’s correlation coefficient. The neural RDM was constructed by calculating the dissimilarity of voxel-wise parametric estimations within brain regions between paths. The Spearman’s rank correlation coefficient (ρ) was then computed between the theoretical RDM and the neural RDM. **(d)** Brain regions identified through voxel-wise searchlight RSA in the whole-brain. Significant regions indicate improved representation on destinations after learning. The color bar represents the Z -values. Abbreviations: HIP, hippocampus; EC, entorhinal cortex; OFC, orbitofrontal cortex; FP, frontal pole; ACC, anterior cingulate cortex; LOC, lateral occipital cortex; TP, temporal pole; V1, primary visual cortex.

Q2. I do not believe that your analyses address your hypothesis, “...if the HIP, EC, and OFC show different activation strengths and representation patterns between two learning stages, they are involved in learning the structure of abstract spaces”. While you do show results in support of the first part of the hypothesis, there is no analysis linking the fMRI data directly to any kind of learning parameter. I believe the authors show this indirectly, by showing that the earlier and later learning stages are related to better accuracy and reaction times from the LMM models. But there is no direct link of behavior to neural data. I believe this should be easy to do with either the univariate or the multivariate analyses.

Response: Thank you for your valuable feedback. We acknowledge that our current analyses only compared brain activation between the exploration and exploitation stages, which is insufficient to draw conclusions about the brain regions involved in learning the structure of abstract spaces. In order to address this concern, we have conducted two additional GLM analyses. Specifically, GLM2 examines the brain activation associated with learning level, and GLM3 examines the brain activation associated with response accuracy. These analyses provide a direct link between the fMRI data and the learning parameters, as suggested. We appreciate your suggestion and have incorporated these additional analyses into our study.

We provided a detailed description of these GLM analyses on page 21 as follows:

“GLM2 was conducted to examine the association between the brain activation and the learning level predicted by the DNN. In the subject-level analysis, we included 17 regressors for each of the five task-fMRI datasets, including 10 regressors for the ten different paths, 1 for the feedback, and 6 nuisance regressors for head motion parameters. The estimated parameter for each path was contrasted to the baseline to obtain a COPE map for the path. Next, we loaded the COPE maps of the paths, set learning level as a regressor, and applied a fixed effects model to estimate the brain activation associated with the learning level. Finally, in the group-level analysis, we used a random effects model to identify brain regions with significant activation associative with the learning level. In addition, we tested the effect of spatial structure on learning by comparing brain activation associated with the learning level across the three different dimensionalities.

GLM3 was set up to detect the brain activation associative with the response

accuracy of each step (*RA*). In the subject-level GLM3 analysis, we set 9 regressors for each of the five task-fMRI datasets, including 1 regressor for the navigation, 1 for the feedback, 1 for *RA* on the navigation period, and 6 nuisance regressors for head motion parameters. Next, we averaged the effect of *RA* on brain activation across the five datasets using a fixed effects model. Finally, in the group-level analysis, we used a random effects model to identify the brain region with significant activation associative with *RA*.”

We present the results of these analyses on page 26, as follows:

“From the GLM2 analysis, we identified brain regions with activity significantly associated with the learning level, which corroborated the findings from the GLM1 analysis (Fig. S9, Supplementary Materials). We detected 11 clusters showing positive association with the learning level. These clusters encompassed the right HIP (28, -46, -8), bilateral MiFG (left: -38, 30, 12; right: 24, 6, 46), bilateral insula (left: -26, 6, 0; right: 30, 20, -8), left SFG (-18, -6, 50), left MeFG (-6, 12, 46), left postcentral gyrus (PoCG) (-60, -28, 40), right IPL (56, -36, 18), and bilateral cerebellum (left: -16, -38, -48; right: 18, -76, -48). Additionally, we identified two clusters in the left MeFG (-2, 58, 10) and right PCC (4, -50, 24), with negative association with the learning level. The detailed information about these clusters is listed in Table S5 (Supplementary Materials). No brain region showed significantly different learning-associative activation across the three dimensionalities.

From the GLM3 analysis, we observed seven clusters showing positive association with *RA* (Fig. 4b). These clusters were located in the right lingual gyrus (20, -84, 4), bilateral MiFG (left: -38, -6, 50; right: 28, -2, 48), bilateral IFG (left: -46, 4, 32; right: 48, 10, 30), right thalamus (24, -26, -2), and left HIP (-20, -32, -2). In addition, we detected nine clusters showing negative association with *RA*. These clusters encompassed the left ACC (4, 56, 4),

bilateral IPL (left: 56, -28, 26; right: -66, -28, 22), right PCC (12, -30, 44), left retrosplenial cortex (RSC) (-12, -58, 14), bilateral middle temporal gyrus (MTG) (left: -58, -20, -10; right: 54, -2, -14), right superior temporal gyrus (STG) (50, -62, 30), and right parahippocampal gyrus (PHG) (26, -48, 12). The detailed information about these clusters is listed in Table S6 (Supplementary Materials).”

Q3. In the discussion, on page 36-37, the authors make a claim about how the anterior and posterior hippocampus show different levels of representation of the two stages, and made a claim that this may indicated specialization. I believe this claim could be support by the data, but the authors did not complete that analysis. A simple analysis to test whether the anterior and posterior hippocampus have significantly different levels of representing learning stages could support their claim.

Response: Thank you for your valuable suggestions. To examine the potential differences in activation or representation levels between the anterior and posterior hippocampus (aHIP and pHIP) for the two learning stages, we conducted both a ROI-based univariate analysis and a ROI-based representational similarity analysis (RSA). For the ROI-based univariate analysis, we extracted the mean signal from each ROI (aHIP and pHIP in both left and right hemispheres), defined according to Howard et al. (2014), separately for the exploration and exploitation stages in the GLM1 analysis. We then performed separate one-way ANOVA tests for each stage to examine the differences among the four ROIs. Our findings showed that in both stages, the activation of pHIP was significantly higher than that of aHIP. To examine

goal representation in each ROI for the exploration and exploitation stages, we conducted a ROI-based RSA. The theoretical representational dissimilarity matrix (RDM) was constructed based on the Euclidean distance between the goals of different paths, while the neural RDM was constructed based on the Pearson's correlation between the COPE maps of the paths. We conducted a repeated measures one-way ANOVA for both the exploration and exploitation stages to examine the differences in goal representations among the four ROIs. However, our analysis did not show any significant differences in goal representation across the subregions.

We have added corresponding description of the ROI-based analyses and results in the Supplementary Materials as follows:

“ROI-based analyses

To explore whether the anterior and posterior hippocampus (aHIP and pHIP) showed different levels of activation or representation in the exploration and exploitation stages, we conducted a ROI-based univariate analysis and a ROI-based representational similarity analysis (RSA).

Regions of interest (ROIs) selection

We defined four subregions of the hippocampus (HIP) based on Howard et al. (2014), which were the left aHIP, left pHIP, right aHIP, and right pHIP. First, the HIP was delineated using the Harvard-Oxford Subcortical Structural Atlas (Desikan et al., 2006), with a probability threshold > 50%. Second, in each brain hemisphere, we divided the HIP into three parts along its long axis, designating the first part (close to the forehead) as aHIP, and the last part (close to the occiput) as pHIP (Fig. S10).

ROI-based univariate analysis

The mean signal of each ROI was extracted separately from the parameter estimation for the exploration and exploitation stages in the GLM1 analysis. One-way repeated measures ANOVA tests were performed separately for the exploration and exploitation stages to examine differences in activation among the four ROIs.

We found significant differences in activity among the HIP subregions both in the exploration stage ($F_{(3, 22)} = 23.38, p < .001$) and in the exploitation stage ($F_{(3, 22)} = 25.04, p < .001$). Specifically, post hoc tests showed that the activity of the left aHIP was significantly lower than that of the left pHIP ($p < .001$ for both stages) and right pHIP ($p < .001$ for both stages). Additionally, the activity of the right aHIP was significantly lower than that of the left pHIP ($p = .001$ for the pre-learning stage and $p < .001$ for the post-learning stage) and right pHIP ($p < .001$ for both stages). The detailed information is listed in Tables S8 and S9.

ROI-based representational similarity analysis (RSA)

An ROI-based RSA was conducted to test whether the four HIP subregions showed different representations to the destinations in the exploration and exploitation stages. For each subject, we compared the theoretical representational dissimilarity matrix (RDM) with the neural RDM for each ROI. First, we constructed a theoretical RDM for each subject by calculating the Euclidean distance between the destinations of different navigational paths (Fig. 4c). Second, we estimated a neural RDM for each subject by calculating the dissimilarity ($1 - r$) between the parameter estimation (PE) maps of each path using NeuroRA (Lu & Ku, 2020), a Python toolbox designed for performing RSA. Third, we compared the neural RDM with the theoretical RDM for each subject using Spearman's rank correlation (ρ). One-way repeated measures

ANOVA tests were conducted separately for the exploration and exploitation stages to examine differences in destination representations among the four ROIs.

We found no significant differences in destination representation among the four subregions of hippocampus, either in the exploration stage ($F_{(3, 22)} = 1.52, p = .237$) or in the exploitation stage ($F_{(3, 22)} = 0.99, p = .417$). The detailed information is listed in Table S8.”

Q4. There should be more justification for why the RSA had to be done. Why is showing a pattern analysis difference unique from showing a magnitude response difference such as the univariate analysis. Perhaps the authors wanted to utilize multi-voxel pattern analysis techniques because it can be more sensitive, particularly to dimensional concepts. Further, differences in the whole-brain univariate and RSA analyses should be highlighted if the RSA did reveal something that for some theoretical reason wouldn't be revealed from the univariate analysis.

a. Chikazoe, J., Lee, D.H., Kriegeskorte, N., Anderson, A.K. (2014). Population coding of affect across stimuli, modalities and individuals. *Nature Neuroscience*, 17(8), 1114–22. <https://doi.org/10.1038/nn.3749>.

b. Popal, H., Wang, Y., Olson, I.R., 2019. A guide to representational similarity analysis for social neuroscience. *Social Cogn. Affect. Neurosci.* 14(11), 1243–1253. <https://doi.org/10.1093/scan/nsz099>.

Response: Thank you for your valuable suggestions. We have revised our study on cognitive map formation and further justified the use of representational similarity

analysis (RSA) method. We aimed to examine whether brain regions develop more accurate representation of the navigational destinations in the exploitation stage than the exploration stage, thereby forming a cognitive map. To address this, we used RSA to assess the neural similarity in relation to the distance between goals across different paths. This allowed us to evaluate the brain regions involved in the formation of a cognitive map within an abstract space.

We chose to use RSA instead of a univariate analysis because the goals varied across different paths, and the analysis required comparisons between paths. It was challenging to achieve this using a univariate analysis, but RSA provided a suitable approach. Moreover, the whole-brain univariate analysis and RSA serve complementary purposes. While the univariate analysis can reveal magnitude response differences, RSA allows us to detect pattern analysis differences that may not be evident through traditional univariate approaches. This approach enhances our understanding of the cognitive processes underlying goal representation and cognitive map formation.

Through RSA, we identified that the orbitofrontal cortex (OFC), frontal pole, entorhinal cortex, and hippocampus showed more accurate representation of destinations in the exploitation stage than the exploration stage. We believe that the inclusion of RSA in our study contributes to a more comprehensive understanding of the cognitive processes underlying goal representation and cognitive map formation.

We have revised the description of the RSA on page 22 as follows:

“A voxel-wise whole-brain searchlight RSA was performed separately for each stage to test whether the brain regions encoded a more accurate representation of destinations in the exploitation stage than the exploration stage. The RSA involved comparing the theoretical representational dissimilarity matrix (RDM) with the neural RDM for each brain region. First, for each subject in both stages, we constructed a theoretical RDM based on the Euclidean distance between the destinations of different paths (Fig. 4c). Second, for a given brain region, we calculated a neural RDM for each subject in both stages. To achieve this, we conducted the subject-level GLM2 analysis on the unsmoothed preprocessed fMRI data, resulting in a parameter estimation (PE) map for each path. We then used NeuroRA (Lu & Ku, 2020), a Python toolbox designed for performing RSA, to calculate the dissimilarity ($1 - r$) between the PE maps of each path, obtaining the neural RDM. Third, we computed Spearman’s rank correlation (ρ) between the neural RDM and the theoretical RDM for each subject. The searchlight size was set to a 3-voxel cube, with a stride of 1 voxel along each axis (x , y , and z). For each stage, we obtained a whole brain ρ -map representing the correlation between the neural RDM and the theoretical RDM for each subject through searchlight RSA. After applying Fisher’s transformation to standardize the ρ -maps, we compared the ρ -maps between the two stages to identify brain regions that showed a more accurate representation of destinations in the exploitation stage than the exploration stage.”

Correspondingly, the results have been updated on page 27 as follows:

“Using whole-brain searchlight RSA, we identified 14 clusters that showed a significantly more accurate representation of destinations in the exploitation stage than the exploration stage (Fig. 4d). These clusters were located in the bilateral entorhinal cortex (left: -16, -2, -22; right: 34, -2, -34), bilateral cuneus

(left: -4, -68, 8; right: 8, -86, 34), left inferior temporal gyrus (ITG) (-54, -20, -36), right ACC (22, 34, 6), left frontal pole (-16, 64, 18), bilateral SFG (left: -8, 54, 40; right: 22, 38, 24), bilateral OFC (left: -4, 70, -8; right: 6, 36, -30), left IFG (-30, 32, -22), and right cerebellum (42, -82, -30). The detailed information about these clusters is listed in Table S7 (Supplementary Materials).”

Q5. How do the authors findings map onto other forms of cognitive maps? Do these results speak to any kind of domain general ability of the medial temporal lobe system and OFC to create cognitive maps? Are these results similar to cognitive map studies showing more abstract knowledge maps? Are these results relevant to spatial navigation or understanding social networks?

Response: Thank you for your valuable feedback. The abstract spaces utilized in our study encompassed both spatial and non-spatial domains. First, they preserved the Euclidean relationships (distances) between different items, which pertains to the spatial domain. Second, the transitions between items were discrete, requiring subjects to infer associations between them, thereby engaging non-spatial cognitive processes. Thus, our study allowed us to investigate the brain regions involved in cognitive map formation, potentially generalizing to both spatial and general cognitive domains.

Similar findings have been reported in both spatial and cognitive domains (Nieh et al., 2021). For example, Patai et al. (2019) examined participants’ navigation in familiar and recently learned physical environments, revealing distinct activations in the posterior hippocampus and retrosplenial cortex (RSC) when navigating these different

environments. Wanjia et al. (2021) demonstrated that the hippocampus formed distinct representations for visually similar stimuli after learning their diverse underlying implications. In our study, we found that both the hippocampus and OFC were jointly involved in learning the structure of abstract spaces, with the hippocampus showing greater engagement in the exploration stage and the OFC in the exploitation stage. This aligns with our recent findings (Liao et al., 2023), where we observed that the hippocampus played a role in gathering and updating information, while the OFC was involved in representing social relationships.

These collective results suggest the involvement of the hippocampus and OFC in forming cognitive maps across spatial and non-spatial domains. Our findings contribute to the understanding of how these brain regions support spatial navigation, abstract knowledge mapping, and potentially extend to the comprehension of social networks.

We have added comparisons between our results and those of previous studies in spatial, cognitive, and social spaces in Discussion, on page 30 as follows:

“Brain activation related to the learning

From GLM1 analysis, we observed distinct brain regions engaged in the exploration and exploitation stages (Fig. 4a). In the exploration, the HIP, lateral PFC, insula, thalami, IPL, and visual cortex showed significantly stronger activation than in the exploitation. These regions likely form a network for mapping abstract spaces during the exploration. Previous studies highlighted the complementary roles of the PFC and HIP in learning and inference (Chaaya

et al., 2018; Lisman et al., 2017; Samborska et al., 2022; Sekeres et al., 2018) and constructing cognitive maps (Costa et al., 2023; Wanjia et al., 2021; Whittington et al., 2022). Additionally, the insula, visual cortex, thalamus, and IPL were implicated in various learning processes (Bick et al., 2019; Chiu et al., 2017; Goold & Meng, 2017; Russ et al., 2003) and coupled with the PFC and HIP in explicit memory (Aggleton et al., 2009; Dickerson et al., 2007; Lech et al., 2016).

In the exploitation, we observed stronger activation in the bilateral mPFC, OFC, and RSC than in the exploration (Fig. 5a). In the exploitation stage, individuals primarily need to compare sensory inputs to the memory retrieved based on the formed map of abstract spaces. The mPFC is a higher-level brain region associated with outcome evaluation, goal planning, action execution, and event prediction (W. H. Alexander & Brown, 2014; Tan et al., 2021). The OFC is involved in inference and formation of cognitive maps in abstract spaces (Niv, 2019; Wikenheiser & Schoenbaum, 2016), while the RSC plays a major role in anchoring the cognitive map in mind to the actual environment and facilitating perspective shifts (A. S. Alexander & Nitz, 2015, 2017). Furthermore, we found that the brain regions significantly activated in the exploration and exploitation stages, including the HIP, OFC, and RSC, were associative with the learning level predicted by the DNN (GLM2, Fig. S9). This result further supports their involvement in forming cognitive map in abstract spaces.

From GLM3 analysis, we also observed that activation in the HIP, lateral PFC, and visual cortex positively correlated with response accuracy, while activation in the OFC and RSC showed a negative correlation with response accuracy (Fig. 4b). The HIP has been implicated in representing distance to goals in both physical and abstract spaces (Crivelli-Decker et al., 2023; Howard et al., 2014; Spiers et al., 2018; Theves et al., 2019). The visual cortex, along with the PHG, is involved in estimating distance (Liang et al., 2023). The OFC is necessary

for forming appropriate behavior-outcome associations (Walton et al., 2010).

The HIP and OFC have been proposed as core regions for encoding cognitive maps (Wikenheiser & Schoenbaum, 2016). From three GLM analyses (Fig. 4), we found different activation patterns of the HIP and OFC. The HIP exhibited greater engagement in the exploration stage or a lower learning level, and its activation was positively associated with behavioral performance. The OFC showed greater engagement in the exploitation stage or a higher learning level, and its activation was negatively associated with behavioral performance.

These results indicated that these two regions may adopt different roles in forming and representing a cognitive map. Previous studies suggested that the HIP and OFC are involved in different aspects of a cognitive map (Liao et al., 2023; Riceberg et al., 2022; Wikenheiser & Schoenbaum, 2016). Our results indicated that the HIP may be responsible for updating the map of spaces and the OFC for utilizing the map to make inferences. In our recent study (Liao et al., 2023), we also found that the HIP was involved in collecting and updating information, whereas the OFC was associated with the relationships between agents in the environment.

Destination representations obtained using RSA on the fMRI data

During the goal-directed navigation in abstract spaces, we found that several brain regions, including the HIP, entorhinal cortex, OFC, and LOC, showed more accurate destination representations in the exploitation stage than the exploration stage (Fig. 4d). When subjects navigated to destinations of nearby locations, these brain regions displayed more similar activation patterns in the exploitation stage. The HIP and entorhinal cortex are well-established central regions for cognitive map-based navigation (Crivelli-Decker et al., 2023; Knudsen & Wallis, 2021; Rueckemann et al., 2021). The OFC was also found representing the future goals during spatial navigation (Basu et al., 2021), and

cooperating with the HIP in forming cognitive maps (Elliott Wimmer & Büchel, 2019; Riceberg et al., 2022). The LOC was suggested as the region for representing object identity, encoding perceptual information of visual stimuli, and coding object interaction (Graumann et al., 2022; Kim & Zatorre, 2011; Roth & Zohary, 2015). Though the LOC was not a typical brain region of cognitive map, previous studies also found that the LOC was essential in processing boundaries and navigational affordances (Bonner & Epstein, 2017; Epstein et al., 2017; Julian et al., 2016). These regions may be involved in encoding and representing spatial information, including the locations of destinations. ”

Reviewer #2:

The authors utilized fMRI to examine how people construct abstract maps. They designed navigation tasks in 1D, 2D, and 3D abstract spaces and acquired fMRI data while subjects explored an abstract space. The authors first employed a deep neural network to capture the behavioral path pattern in the navigation and used a k-means algorithm to separate the task into pre- and post-learning stages. Then they compared the brain activations and representations between the pre- and post-learning stages. They found significantly different activations between the two stages that included the bilateral hippocampus, lateral and medial prefrontal cortex, and precuneus. The hippocampus also showed different representations of the paths of the two stages.

Overall, I am not convinced that the claims made are supported by the data.

Throughout, I found the data presentation confusing and unclear.

Response: Thank you for dedicating your time and effort to carefully review our work. We sincerely apologize for any confusion or lack of clarity in the manuscript. Based on your valuable feedback, we have made significant revisions to address these issues. Specifically, we have provided clearer explanations regarding the use of the deep neural network (DNN) as a preparation for the fMRI data analysis. Additionally, we have conducted further analyses to examine whether the subjects formed improved metric representations after learning, addressing one of your concerns.

Q1. First, I don't understand why DNN was used to analyze behavioral data. The authors convert the response accuracy (RAstep) and response time (RTstep) on a path into a 2-dimensional matrix and give it as input to a classifier. However, DNN itself is a black box, and this analysis still doesn't tell us which behavioral features change fundamentally, compared to fig2b.

Response: Thank you for raising this question. The purpose of using the DNN in the analysis of behavioral data was to estimate the level to which the subjects had acquired the spatial structure in each path. The acquisition of the spatial structure in abstract spaces is a complex process, and it is challenging to determine the level of understanding a subject has reached during navigation. By employing a DNN, which is a powerful machine learning algorithm capable of capturing subtle patterns in input data through layered neural networks, we aimed to capture the underlying structure of the subjects' navigation behavior. While the DNN itself may be considered a black box that does not explicitly identify the specific behavioral features that undergo

fundamental changes, it provides a valuable tool for investigating the brain activation and representation associated with learning during navigation in abstract spaces.

We have revised the description regarding our use of DNN on page 16 as follows:

The RA_{step} or RT_{step} in each path were combined into a $N_{step} \times 2$ matrix \mathbf{B} .

$$\mathbf{B} = \begin{bmatrix} RA_1 & RT_1 \\ \vdots & \vdots \\ RA_{N_{step}} & RT_{N_{step}} \end{bmatrix}$$

We developed a DNN prediction model to estimate the subjects' learning level of the abstract spatial structure based on the \mathbf{B} matrix. DNN is a powerful machine learning algorithm that can capture subtle characteristics in the input data by stacking layers of neural networks (Schmidhuber, 2015). We first used the paths of the early learning and late learning phases to train the DNN, enabling it to learn and recognize the distinctive features of each phase. Subsequently, we used the trained DNN to estimate the probability of each path belonging to either the early or late learning phase (Eq. 3), which was defined as the learning level of the abstract space.

In addition, we have revised Fig. 3 trying to clarify the application methods.

Fig. 3. Schematic and results for the separation of the navigation paths into the exploration and exploitation stages. **(a)** Definition of the early learning (green), mid-learning (white), and late learning (orange). The first three paths and the last three paths of each space were labeled as the early and late learning phases, respectively. These labeled paths were used in training the deep neural network (DNN). **(b)** Construction of the DNN prediction model. The DNN contains 2 hidden layers between the input and output layers. The input data were a N_{step} -by-2 matrix $\mathbf{B} = [RA_1, RT_1; \dots; RA_{N_{step}}, RT_{N_{step}}]$ of a path, where $RA_{N_{step}}$ and $RT_{N_{step}}$ represent response accuracy and response time of the N^{th} step, respectively. The values of the two units in the output layer indicate the probability of the path being categorized as early learning phase (denoted as $P(early)$) and being categorized as late

learning phase (denoted as $P(\text{late}) = 1 - P(\text{early})$). The **B** matrices of all paths were input to the trained DNN to obtain the $P(\text{early})$. **(c)** Categorization the navigation paths into the exploration and exploitation stages. A k -means algorithm was used to categorize the paths into the exploration and exploitation stages based on the $P(\text{early})$. **(d)** Performance of the DNN prediction model. During training, the DNN model predicted the label of the paths in the test set. The accuracy was calculated as the ratio of correctly predicted paths to the total paths in the testing set and averaged across the 100 iterations. The model accuracy value was significantly higher than the chance level ($0.25 = 0.5 / n_{\text{test}}$). **(e)** The number of paths categorized as exploitation stage by the k -means algorithm in each space. ***, $p < .001$.

Q2. Second, the authors just analyze the learning effect, which is present in many experimental tasks. They compared activation in early and late stages and therefore argued that this change reflects which brain regions construct cognitive maps. Obviously, this simple analysis cannot support the conclusion. The key to the construction of cognitive maps lies in how to learn the connections between different stimuli, for example, using RSA to analyze whether the neural similarities between different stimuli conform to the metric features of cognitive maps or to reveal the neural geometric structures between stimuli. Then to analyze how these neural geometric structures evolve with learning.

Response: Thank you for your insightful input. We acknowledge that solely comparing brain activation between the exploration and exploitation stages is insufficient to support our conclusion. Taking your suggestion into consideration, we conducted an RSA to examine whether the subjects developed more accurate

representations of the navigational goals in the exploitation stage than the exploration stage. Specifically, we constructed theoretical representation dissimilarity matrices (RDMs) based on the Euclidean distance between the goals of the paths. Additionally, we constructed neural RDMs by calculating the dissimilarity $(1 - r)$ between the parametric estimation maps of the paths. Subsequently, we separately compared the neural RDMs and theoretical RDMs for the exploration and exploitation stages. Finally, we compared the RSA results between the two stages to identify the brain regions that exhibited differential representation of navigational goals during the two learning stages.

We have revised the description of the RSA on page 22 as follows:

“A voxel-wise whole-brain searchlight RSA was performed separately for each stage to test whether the brain regions encoded a more accurate representation of destinations in the exploitation stage than the exploration stage. The RSA involved comparing the theoretical representational dissimilarity matrix (RDM) with the neural RDM for each brain region. First, for each subject in both stages, we constructed a theoretical RDM based on the Euclidean distance between the destinations of different paths (Fig. 4c). Second, for a given brain region, we calculated a neural RDM for each subject in both stages. To achieve this, we conducted the subject-level GLM2 analysis on the unsmoothed preprocessed fMRI data, resulting in a parameter estimation (PE) map for each path. We then used NeuroRA (Lu & Ku, 2020), a Python toolbox designed for performing RSA, to calculate the dissimilarity $(1 - r)$ between the PE maps of each path, obtaining the neural RDM. Third, we computed Spearman’s rank correlation (ρ) between the neural RDM and the theoretical RDM for each subject. The searchlight size was set to a 3-voxel cube, with a stride of 1 voxel

along each axis (x , y , and z). For each stage, we obtained a whole brain ρ -map representing the correlation between the neural RDM and the theoretical RDM for each subject through searchlight RSA. After applying Fisher's transformation to standardize the ρ -maps, we compared the ρ -maps between the two stages to identify brain regions that showed a more accurate representation of destinations in the exploitation stage than the exploration stage.”

Correspondingly, the results have been updated on page 27 as follows:

“Using whole-brain searchlight RSA, we identified 14 clusters that showed a significantly more accurate representation of destinations in the exploitation stage than the exploration stage (Fig. 4d). These clusters were located in the bilateral entorhinal cortex (left: -16, -2, -22; right: 34, -2, -34), bilateral cuneus (left: -4, -68, 8; right: 8, -86, 34), left inferior temporal gyrus (ITG) (-54, -20, -36), right ACC (22, 34, 6), left frontal pole (-16, 64, 18), bilateral SFG (left: -8, 54, 40; right: 22, 38, 24), bilateral OFC (left: -4, 70, -8; right: 6, 36, -30), left IFG (-30, 32, -22), and right cerebellum (42, -82, -30). The detailed information about these clusters is listed in Table S7 (Supplementary Materials).”

Fig. 4. Brain regions associated with learning in abstract spaces. **(a)** Brain regions showing significant differences in activation between the exploration and exploitation stages. Warm (cold) colors stand for the contrast of “exploration > exploitation” (“exploration < exploitation”). **(b)** Brain regions with the activation associated with response accuracy. Warm (cold) colors stand for a positive (negative) association. The color bar represents the Z-values. **(c)** Schematic for the representational similarity analysis (RSA). The lower triangular matrix depicts a theoretical representational dissimilarity matrix (RDM) where each element was the Euclidean distance (d) between goals of different paths. The upper triangular matrix depicts a neural RDM, with each element was the dissimilarity ($1 - r$) between two paths, where r is the Pearson’s correlation coefficient. The neural RDM was constructed by calculating the dissimilarity of voxel-wise parametric estimations within brain regions between paths. The Spearman’s rank correlation coefficient (ρ) was then computed between the theoretical RDM and the neural RDM. **(d)** Brain regions identified through voxel-wise searchlight RSA in the whole-brain. Significant regions indicate improved

representation on destinations after learning. The color bar represents the Z -values. Abbreviations: HIP, hippocampus; EC, entorhinal cortex; OFC, orbitofrontal cortex; FP, frontal pole; ACC, anterior cingulate cortex; LOC, lateral occipital cortex; TP, temporal pole; V1, primary visual cortex.

Q3. Third, most studies of abstract space involve the integration of different dimensions, that is, the linkages between stimuli. In this experiment, however, subjects did not seem to need to integrate different dimensions. Therefore, I do not think this is an elegant design for studying abstract space construction. The decision at each step can be seen as a simple value decision.

Response: Thank you for your valuable comment. We agree with you that our experiment design did not strictly require the subjects to integrate different dimensions. In fact, we took several measures to encourage the integration of multiple dimensional information. First, the visual stimuli used in the experiment were intentionally designed to be visually similar, with options arranged based on their location within the abstract spaces. This prompted subjects to base their responses on the structure of the abstract spaces rather than relying solely on visual features. We hypothesized that the learning process may initially involve simple value decisions and gradually transition towards decisions based on the abstract space's structure. Second, the subjects were asked to collect the goals with minimal steps to avoid a 20% probability of forfeiting the goal's reward. This incentivized subjects to integrate different dimensional information instead of considering each dimension separately. Third, in our analyses, we compared brain activations and representations between the

exploration and exploitation stages, which helped mitigate the potential influence of visual features alone on our findings, as both stages involved similar visual stimuli.

Nevertheless, we have to admit that these measures may not fully control for the possibility that the subjects considered the different dimensional information separately across the task. We have added a statement on page 33, line 21, to address this limitation:

“We took measures to encourage the subjects integrating different dimensional information during making decisions. Nevertheless, we cannot rule out the possibility that the subjects may consider the information of different dimensions separately.”

References:

- Aggleton, J. P., Poirier, G. L., Aggleton, H. S., Vann, S. D., & Pearce, J. M. (2009). Lesions of the fornix and anterior thalamic nuclei dissociate different aspects of hippocampal-dependent spatial learning: Implications for the neural basis of scene learning. *Behavioral Neuroscience*, *123*(3), 504–519. <https://doi.org/10.1037/a0015404>
- Alexander, A. S., & Nitz, D. A. (2015). Retrosplenial cortex maps the conjunction of internal and external spaces. *Nature Neuroscience*, *18*(8), 1143–1151. <https://doi.org/10.1038/nn.4058>
- Alexander, A. S., & Nitz, D. A. (2017). Spatially Periodic Activation Patterns of Retrosplenial Cortex Encode Route Sub-spaces and Distance Traveled. *Current Biology*, *27*(11), 1551-1560.e4. <https://doi.org/10.1016/j.cub.2017.04.036>
- Alexander, W. H., & Brown, J. W. (2014). A general role for medial prefrontal cortex in event prediction. *Frontiers in Computational Neuroscience*, *8*. <https://www.frontiersin.org/articles/10.3389/fncom.2014.00069>
- Basu, R., Gebauer, R., Herfurth, T., Kolb, S., Golipour, Z., Tchumatchenko, T., & Ito, H. T. (2021). The orbitofrontal cortex maps future navigational goals. *NATURE*, *599*(7885), 449–452. <https://doi.org/10.1038/s41586-021-04042-9>
- Bick, S. K., Patel, S. R., Katnani, H. A., Peled, N., Widge, A., Cash, S. S., & Eskandar, E. N. (2019). Caudate stimulation enhances learning. *Brain*, *142*(10), 2930–2937. <https://doi.org/10.1093/brain/awz254>
- Bonner, M. F., & Epstein, R. A. (2017). Coding of navigational affordances in the human visual system. *Proceedings of the National Academy of Sciences*, *114*(18), 4793–4798. <https://doi.org/10.1073/pnas.1618228114>
- Chaaya, N., Battle, A. R., & Johnson, L. R. (2018). An update on contextual fear memory mechanisms: Transition between Amygdala and Hippocampus. *Neuroscience & Biobehavioral Reviews*, *92*, 43–54. <https://doi.org/10.1016/j.neubiorev.2018.05.013>
- Chiu, Y.-C., Jiang, J., & Egner, T. (2017). The Caudate Nucleus Mediates Learning of Stimulus–Control State Associations. *The Journal of Neuroscience*, *37*(4), 1028–1038. <https://doi.org/10.1523/JNEUROSCI.0778-16.2016>
- Costa, K. M., Scholz, R., Lloyd, K., Moreno-Castilla, P., Gardner, M. P. H., Dayan, P., & Schoenbaum, G. (2023). The role of the lateral orbitofrontal cortex in creating cognitive maps. *Nature Neuroscience*, *26*(1), 107–115. <https://doi.org/10.1038/s41593-022-01216-0>

- Crivelli-Decker, J., Clarke, A., Park, S. A., Huffman, D. J., Boorman, E. D., & Ranganath, C. (2023). Goal-oriented representations in the human hippocampus during planning and navigation. *Nature Communications*, *14*(1), 2946. <https://doi.org/10.1038/s41467-023-35967-6>
- Desikan, R. S., Segonne, F., Fischl, B., Quinn, B. T., Dickerson, B. C., Blacker, D., Buckner, R. L., Dale, A. M., Maguire, R. P., Hyman, B. T., Albert, M. S., & Killiany, R. J. (2006). An automated labeling system for subdividing the human cerebral cortex on MRI scans into gyral based regions of interest. *Neuroimage*, *31*(3), 968–980. <https://doi.org/10.1016/j.neuroimage.2006.01.021>
- Dickerson, B. C., Miller, S. L., Greve, D. N., Dale, A. M., Albert, M. S., Schacter, D. L., & Sperling, R. A. (2007). Prefrontal-hippocampal-fusiform activity during encoding predicts intraindividual differences in free recall ability: An event-related functional-anatomic MRI study. *Hippocampus*, *17*(11), 1060–1070. <https://doi.org/10.1002/hipo.20338>
- Elliott Wimmer, G., & Büchel, C. (2019). Learning of distant state predictions by the orbitofrontal cortex in humans. *Nature Communications*, *10*(1), 2554. <https://doi.org/10.1038/s41467-019-10597-z>
- Epstein, R. A., Patai, E. Z., Julian, J. B., & Spiers, H. J. (2017). The cognitive map in humans: Spatial navigation and beyond. *Nature Neuroscience*, *20*(11), 1504–1513. <https://doi.org/10.1038/nn.4656>
- Goold, J. E., & Meng, M. (2017). Categorical learning revealed in activity pattern of left fusiform cortex. *Human Brain Mapping*, *38*(7), 3648–3658. <https://doi.org/10.1002/hbm.23620>
- Graumann, M., Ciuffi, C., Dwivedi, K., Roig, G., & Cichy, R. M. (2022). The spatiotemporal neural dynamics of object location representations in the human brain. *Nature Human Behaviour*, *6*(6), 796–811. <https://doi.org/10.1038/s41562-022-01302-0>
- Howard, L. R., Javadi, A. H., Yu, Y., Mill, R. D., Morrison, L. C., Knight, R., Loftus, M. M., Staskute, L., & Spiers, H. J. (2014). The Hippocampus and Entorhinal Cortex Encode the Path and Euclidean Distances to Goals during Navigation. *Current Biology*, *24*(12), 1331–1340. <https://doi.org/10.1016/j.cub.2014.05.001>
- Julian, J. B., Ryan, J., Hamilton, R. H., & Epstein, R. A. (2016). The Occipital Place Area Is Causally Involved in Representing Environmental Boundaries during Navigation. *Current Biology*, *26*(8), 1104–1109. <https://doi.org/10.1016/j.cub.2016.02.066>
- Kim, J.-K., & Zatorre, R. J. (2011). Tactile-Auditory Shape Learning Engages the

- Lateral Occipital Complex. *Journal of Neuroscience*, 31(21), 7848–7856.
<https://doi.org/10.1523/JNEUROSCI.3399-10.2011>
- Knudsen, E. B., & Wallis, J. D. (2021). Hippocampal neurons construct a map of an abstract value space. *Cell*, 184(18), 4640–4650.e10.
<https://doi.org/10.1016/j.cell.2021.07.010>
- Lech, R. K., Güntürkün, O., & Suchan, B. (2016). An interplay of fusiform gyrus and hippocampus enables prototype- and exemplar-based category learning. *Behavioural Brain Research*, 311, 239–246. <https://doi.org/10.1016/j.bbr.2016.05.049>
- Liang, Q., Liao, J., Li, J., Zheng, S., Jiang, X., & Huang, R. (2023). The role of the parahippocampal cortex in landmark-based distance estimation based on the contextual hypothesis. *Human Brain Mapping*, 44(1), 131–141.
<https://doi.org/10.1002/hbm.26069>
- Liao, J., Li, J., Qiu, Y., Wu, X., Liu, B., Zhang, L., Zhang, Y., Peng, X., & Huang, R. (2023). Dissociable contributions of the hippocampus and orbitofrontal cortex to representing task space in a social context. *Cerebral Cortex*, bhad447.
<https://doi.org/10.1093/cercor/bhad447>
- Lisman, J., Buzsáki, G., Eichenbaum, H., Nadel, L., Ranganath, C., & Redish, A. D. (2017). Viewpoints: How the hippocampus contributes to memory, navigation and cognition. *Nature Neuroscience*, 20(11), 1434–1447. <https://doi.org/10.1038/nn.4661>
- Lu, Z., & Ku, Y. (2020). NeuroRA: A Python Toolbox of Representational Analysis From Multi-Modal Neural Data. *Frontiers in Neuroinformatics*, 14.
<https://www.frontiersin.org/article/10.3389/fninf.2020.563669>
- Nieh, E. H., Schottdorf, M., Freeman, N. W., Low, R. J., Lewallen, S., Koay, S. A., Pinto, L., Gauthier, J. L., Brody, C. D., & Tank, D. W. (2021). Geometry of abstract learned knowledge in the hippocampus. *Nature*, 595(7865), 80–+.
<https://doi.org/10.1038/s41586-021-03652-7>
- Niv, Y. (2019). Learning task-state representations. *Nature Neuroscience*, 22(10), 1544–1553. <https://doi.org/10.1038/s41593-019-0470-8>
- Patai, E. Z., Javadi, A.-H., Ozubko, J. D., O’Callaghan, A., Ji, S., Robin, J., Grady, C., Winocur, G., Rosenbaum, R. S., Moscovitch, M., & Spiers, H. J. (2019). Hippocampal and Retrosplenial Goal Distance Coding After Long-term Consolidation of a Real-World Environment. *Cerebral Cortex*, 29(6), 2748–2758.
<https://doi.org/10.1093/cercor/bhz044>
- Riceberg, J. S., Srinivasan, A., Guise, K. G., & Shapiro, M. L. (2022). Hippocampal signals modify orbitofrontal representations to learn new paths. *CURRENT BIOLOGY*,

32(15), 3407–3413. <https://doi.org/10.1016/j.cub.2022.06.010>

Roth, Z. N., & Zohary, E. (2015). Fingerprints of Learned Object Recognition Seen in the fMRI Activation Patterns of Lateral Occipital Complex. *Cerebral Cortex*, 25(9), 2427–2439. <https://doi.org/10.1093/cercor/bhu042>

Rueckemann, J. W., Sosa, M., Giocomo, L. M., & Buffalo, E. A. (2021). The grid code for ordered experience. *Nature Reviews Neuroscience*, 22(10), 637–649. <https://doi.org/10.1038/s41583-021-00499-9>

Russ, M. O., Mack, W., Grama, C.-R., Lanfermann, H., & Knopf, M. (2003). Enactment effect in memory: Evidence concerning the function of the supramarginal gyrus. *Experimental Brain Research*, 149(4), 497–504. <https://doi.org/10.1007/s00221-003-1398-4>

Samborska, V., Butler, J. L., Walton, M. E., Behrens, T. E. J., & Akam, T. (2022). Complementary task representations in hippocampus and prefrontal cortex for generalizing the structure of problems. *Nature Neuroscience*, 25(10), Article 10. <https://doi.org/10.1038/s41593-022-01149-8>

Schmidhuber, J. (2015). Deep learning in neural networks: An overview. *Neural Networks*, 61, 85–117. <https://doi.org/10.1016/j.neunet.2014.09.003>

Sekeres, M. J., Winocur, G., & Moscovitch, M. (2018). The hippocampus and related neocortical structures in memory transformation. *Neuroscience Letters*, 680, 39–53. <https://doi.org/10.1016/j.neulet.2018.05.006>

Spiers, H. J., Olafsdottir, H. F., & Lever, C. (2018). Hippocampal CA1 activity correlated with the distance to the goal and navigation performance. *Hippocampus*, 28(9), 644–658. <https://doi.org/10.1002/hipo.22813>

Tan, J., Shen, X., Zhang, X., & Wang, Y. (2021). Multivariate Encoding Analysis of Medial Prefrontal Cortex Cortical Activity during Task Learning. *2021 43rd Annual International Conference of the IEEE Engineering in Medicine & Biology Society (EMBC)*, 6699–6702. <https://doi.org/10.1109/EMBC46164.2021.9630322>

Theves, S., Fernandez, G., & Doeller, C. F. (2019). The Hippocampus Encodes Distances in Multidimensional Feature Space. *Current Biology*, 29(7), 1226-1231.e3. <https://doi.org/10.1016/j.cub.2019.02.035>

Walton, M. E., Behrens, T. E. J., Buckley, M. J., Rudebeck, P. H., & Rushworth, M. F. S. (2010). Separable Learning Systems in the Macaque Brain and the Role of Orbitofrontal Cortex in Contingent Learning. *Neuron*, 65(6), 927–939. <https://doi.org/10.1016/j.neuron.2010.02.027>

Wanjia, G., Favila, S. E., Kim, G., Molitor, R. J., & Kuhl, B. A. (2021). Abrupt hippocampal remapping signals resolution of memory interference. *Nature Communications*, *12*(1), 4816. <https://doi.org/10.1038/s41467-021-25126-0>

Whittington, J. C. R., McCaffary, D., Bakermans, J. J. W., & Behrens, T. E. J. (2022). How to build a cognitive map. *Nature Neuroscience*, *25*(10), 1257–1272. <https://doi.org/10.1038/s41593-022-01153-y>

Wikenheiser, A. M., & Schoenbaum, G. (2016). Over the river, through the woods: Cognitive maps in the hippocampus and orbitofrontal cortex. *Nature Reviews Neuroscience*, *17*(8), 513–523. <https://doi.org/10.1038/nrn.2016.56>

REVIEWERS' COMMENTS:

Reviewer #1 (Remarks to the Author):

Thank you for addressing my comments. The additional detail justifying your analysis is very helpful.

(Editor remarks: R2 was withdrawn and R1 comments on the major points from R2 follow below):

Question 1: I believe DNNs are useful in providing us with information that cannot be directly captured by participants' behaviors in a task. I understand that some in the field do not like this, and have a fundamental objection because they believe it can lead to lazy task design. If that is what reviewer 2 was getting at, I do not believe the author's revisions could have satisfied them. Unfortunately, I do not have the machine learning expertise to weigh in on whether the DNN analysis was appropriate or not. My intuition is that it is adding something interesting to the paper, that I think could be a unique contribution to the literature with a new technique.

Question 2: I believe they have addressed this appropriately. I had a comment about the RSA as well, and I think it is more clear what their objective was and their claim on learning cognitive maps is better supported.

Question 3: Just being very frank here. As a reviewer, I do not believe it is my place to tell the authors what they should have done in their hypothesis testing and experimental design. I believe my job is to judge what they did do, and whether their claims are supported by what they did. So I fundamentally disagree with this reviewer's comment. I think the authors have made it clear what their "stamp" in the field this study is fulfilling have stated the limited scope they are trying to address. For example, I am very interested in the cerebellum and I enjoyed reading their findings reporting on this. But I do not believe I should steer their hypotheses and testing to explore the cerebellum further.

Response to Reviewers' comments

Manuscript Number: COMMSBIO-23-2026B

Manuscript Title: Forming cognitive maps for abstract spaces: the roles of the human hippocampus and orbitofrontal cortex

Reviewer #1:

Thank you for addressing my comments. The additional detail justifying your analysis is very helpful.

Response: Thank you for your valuable feedback and for recognizing the efforts we made to address your previous comments.

Comments on the major points from Reviewer #2

Q1. I believe DNNs are useful in providing us with information that cannot be directly captured by participants' behaviors in a task. I understand that some in the field do not like this, and have a fundamental objection because they believe it can lead to lazy task design. If that is what reviewer 2 was getting at, I do not believe the author's revisions could have satisfied them. Unfortunately, I do not have the machine learning expertise to weigh in on whether the DNN analysis was appropriate or not. My intuition is that it is adding something interesting to the paper, that I think could be a unique contribution to the literature with a new technique.

Response: Thank you for your encouragement for using the DNN analysis in the study. We followed rigorous design and training processes to ensure the validity of the DNN model in the study. We acknowledge that the DNN analysis is a new valuable technique, but it cannot replace the importance of a well-design experiment. We will place greater emphasis on experimental design in future studies.

Q2. I believe they have addressed this appropriately. I had a comment about the RSA as well, and I think it is more clear what their objective was and their claim on learning cognitive maps is better support.

Response: Thank you for your valuable feedback.

Q3. Just being very frank here. As a reviewer, I do not believe it is my place to tell the authors what they should have done in their hypothesis testing and experimental design. I believe my job is to judge what they did do, and whether their claims are supported by what they did. So I fundamentally disagree with this reviewer's comment. I think the authors have made it clear what their "stamp" in the field this study is fulfilling have stated the limited scope they are trying to address. For example, I am very interested in the cerebellum and I enjoyed reading their findings reporting on this. But I do not believe I should steer their hypotheses and testing to explore the cerebellum further.

Response: We sincerely appreciate your valuable insights and support throughout the review process. Your feedback has been instrumental in enhancing the quality of our

manuscript. Your suggestion also inspires us to explore whether the cerebellum is involved in building the abstract cognitive maps in the future studies. Thank you for your time and thoughtful evaluation.